# Self-Steering Language Models

**Gabriel Grand**[1*]   **Joshua B. Tenenbaum**[1]    **Vikash K. Mansinghka**[1]
**Alexander K. Lew**[1,2]   **Jacob Andreas**[1]

[1]Massachusetts Institute of Technology    [2]Yale University
{gg, jbt, vkm, jda}@mit.edu   alexander.lew@yale.edu

 github.com/gabegrand/self-steering

## Abstract

While test-time reasoning enables language models (LMs) to tackle complex tasks, searching or planning in natural language can be slow, costly, and error-prone. But even when LMs struggle to emulate the precise reasoning steps needed to solve a problem, they often excel at describing its *abstract structure*—both how to verify solutions and *how to search* for them. This paper introduces DISCIPL, a method for "self-steering" LMs where a *Planner model* generates a task-specific *inference program* that is executed by a population of *Follower models*. Our approach equips LMs with the ability to write recursive search procedures that guide LM inference, enabling new forms of verifiable and efficient reasoning. When instantiated with a small Follower (e.g., Llama-3.2-1B or Qwen3-1.7B), DISCIPL matches (and sometimes outperforms) much larger models, including GPT-4o and o1, on challenging constrained generation tasks. Our work opens up a design space of highly-parallelized Monte Carlo inference strategies that outperform standard best-of-$N$ sampling, require no finetuning, and can be implemented automatically by existing LMs.

## 1   Introduction

Even as language models (LMs) are becoming increasingly proficient at reasoning tasks, progress has been "jagged" (Karpathy, 2024; Roose, 2025): today's frontier models surpass experts at science and math reasoning (e.g., Hendrycks et al., 2021; Rein et al., 2023; Wang et al., 2024b) yet still routinely struggle with counting, arithmetic, tic-tac-toe, metered poetry, and other intuitively simple tasks (Ball et al., 2024; McCoy et al., 2023; Xu & Ma, 2024). For example, even very capable LMs have difficulty writing a coherent sentence under the constraints in Fig. 1, which are manageable for most proficient English speakers.

There is a growing consensus that many problems require a more deliberate, effortful style of thinking (Kahneman, 2011). However, an open question is how to structure inference so as to best leverage available computational resources. One popular approach performs in-context reasoning via serialized chain-of-thought (DeepSeek-AI et al., 2025; OpenAI, 2024b). While highly flexible, reasoning via autoregressive generation is costly, slow, and can still produce unreliable outputs. On the other hand, structured inference methods like tree search (Silver et al., 2016; Yao et al., 2023) and sequential Monte Carlo (Lew et al., 2023; Loula et al., 2025; Zhao et al., 2024) attain better parallelism and efficiency by coordinating test-time computation via external algorithms. However, these methods require significant hand-engineering and rely on pre-defined scorers or verifiers, limiting their applicability.

In this work, we propose a new meta-reasoning framework called DISCIPL in which language models *themselves* drive the decisions for how to structure inference-time computation. In our approach, a **Planner LM** generates an ad-hoc problem specification (encoding its understanding of the task requirements) and inference procedure (encoding its plan for how to solve the task). Importantly, the specification and plan are implemented as *inference programs* that invoke **Follower LMs**, either generatively or as likelihood evaluators. By decomposing reasoning into planning and execution, our architecture preserves flexibility while enabling orchestration of highly efficient, parallel search patterns.

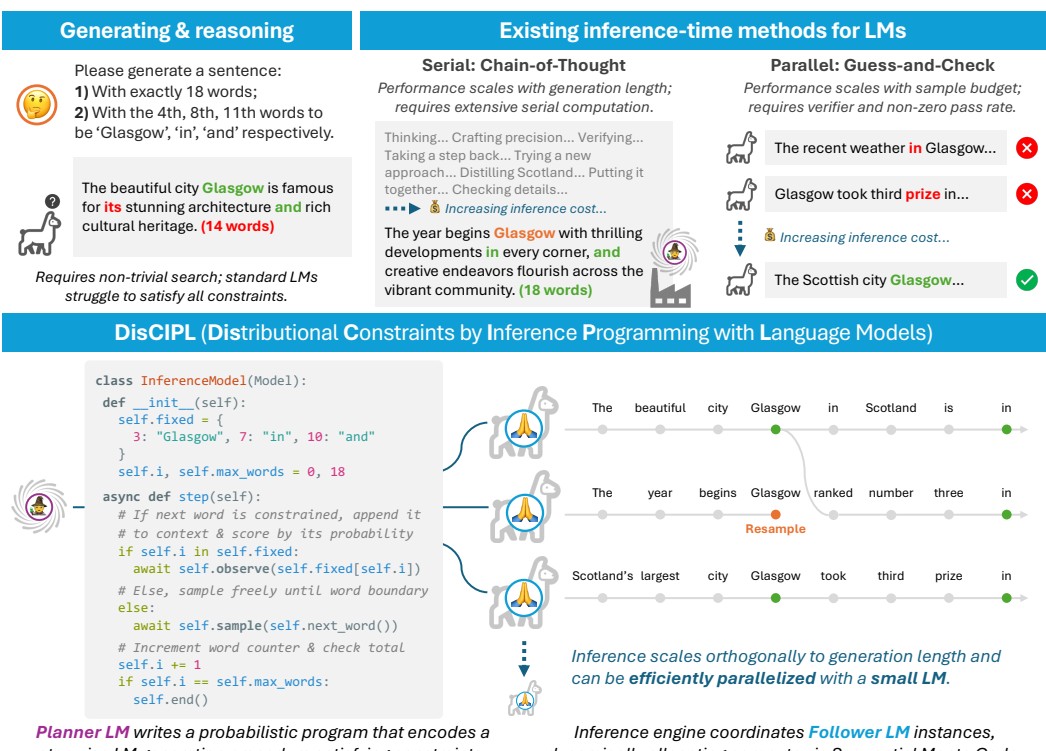

Figure 1: **Self-steering language models with probabilistic programs.** LMs struggle with problems that combine generation and reasoning under constraints (*top left*, task from Yao et al., 2024). Popular approaches (*top right*) scale inference by search or sampling; however, even costly reasoning models do not always yield correct or fluent results (CoT example from o1, single-shot, first attempt). In our method (DISCIPL, *bottom*), a Planner LM writes an *inference program* that defines step-by-step computations to steer a population of Follower LMs. Our approach combines the benefits of serial and parallel methods: the Planner ensures correctness by construction, while the Followers collectively search for sequences with high probability. (See Fig. 7 for a detailed visualization.)

To test this approach, we evaluate DISCIPL on two domains: (1) COLLIE (Yao et al., 2024), a challenging constrained generation benchmark on which even very large LMs perform unreliably; and (2) PUZZLES, a custom dataset of difficult tasks involving poetry composition, grant-writing, budgeting, and itinerary planning. We instantiate DISCIPL using a capable Planner LM to generate code (GPT-4o prompted with few-shot examples), and a small Follower LM to execute it (lightweight versions of Llama-3 or Qwen3). On paragraph tasks, this approach boosts accuracy well beyond the original capabilities of the Follower. Meanwhile, on more tightly constrained sentence and puzzle tasks, DISCIPL enables the Follower to surpass the Planner and approach the performance of powerful reasoning models like o1. Finally, we study how DISCIPL scales with the size and capabilities of the underlying Follower model, allowing our approach to benefit organically from advances in LM architecture and training methods.

Amidst a growing body of task-specific search and sampling methods for LMs (§2), we introduce a generalized approach. By using LMs to implement their own inference procedures on-the-fly, DISCIPL yields efficient solutions to problems that would otherwise require non-trivial human engineering. We believe that this new combination of code generation and probabilistic inference represents an exciting step in test-time scaling.

## 2   Related Work

**Scaling Inference-Time Computation.**  Reasoning via autoregressive generation is effective for many tasks, and the success of "chain-of-thought" (Nye et al., 2021; Wei et al., 2022) has spawned many variants that trade off generality and efficiency: whereas constructing

an explicit "tree-of-thought" (Liu et al., 2023; Yao et al., 2023) requires problem-specific engineering, linearizing reasoning into one long "stream-of-search" (Gandhi et al., 2024; Lehnert et al., 2024) scales exponentially with tree depth. As an alternative, simple sampling procedures like best-of-$N$ sampling (Brown et al., 2024; Cobbe et al., 2021) and self-consistency/majority voting (Wang et al., 2023a) have also emerged as a popular means of scaling LM performance. These "embarrassingly parallel" methods (Herlihy & Shavit, 2012) are simple to implement, but can be computationally wasteful (Damani et al., 2024; Snell et al., 2024). By combining serial and parallel methods, our approach achieves a high degree of resource efficiency while maintaining flexibility.

**Parallel and Asynchronous Generation.** A variety of recent proposals leverage parallel and/or asynchronous generation with LMs (Jin et al., 2025; Ning et al., 2024; Pan et al., 2025; Rodionov et al., 2025). These methods typically define lightweight markup syntax to expose specific operations, like spawning and joining threads. In contrast, our approach offers far greater expressivity by building on a Turing-complete programming language.

**Self-Improvement.** Recently, a variety of novel methods have been proposed for using LMs to optimize prompts (Fernando et al., 2024; Honovich et al., 2023; Khattab et al., 2023; Shinn et al., 2023; Yang et al., 2024; Zhou et al., 2023c), agentic systems (Hu et al., 2024), and optimization procedures themselves (Zelikman et al., 2023). DISCIPL shares a similar recursive flavor, but generates ad-hoc *inference algorithms* that provide token-level control over LM behavior at decoding time with probabilistic guarantees.

**Constrained Generation.** Recent years have seen a proliferation of benchmarks designed to test the ability of LMs to adhere to complex and compositional constraints (Chia et al., 2024; Jiang et al., 2023; Lin et al., 2020; Sun et al., 2023; Wang et al., 2022; Yao et al., 2024; Zhou et al., 2023a). Prior approaches have focused on developing decoding algorithms for specific classes of constraints (Hokamp & Liu, 2017; Koo et al., 2024; Lu et al., 2021; 2022; Poesia et al., 2022; Post & Vilar, 2018; Ugare et al., 2024; Willard & Louf, 2023) or guiding generation with neural models (Amini et al., 2023; Kumar et al., 2022; Li et al., 2022; Qin et al., 2022). Various learning-based approaches have also been proposed, including self-correction through feedback (Welleck et al., 2023) and bootstrapping from self-generated instructions (Wang et al., 2023b; Zhou et al., 2023b).

**Sequential Monte Carlo with LMs.** Particle-based methods, such as sequential Monte Carlo (SMC; Doucet et al., 2001), offer a powerful framework for building adaptive inference-time search procedures. SMC has recently been applied to LMs to solve various tasks, including constrained generation (Lew et al., 2023; Loula et al., 2025; Zhao et al., 2024) and math reasoning (Feng et al., 2024; Puri et al., 2025). However, applying SMC to new problems requires either engineering or learning various parameters of the algorithm (e.g., reward models or twist functions); in our work, an LM automates this process.

**Probabilistic Programming and LMs.** Probabilistic programming languages (PPLs) allow users to implement probabilistic models as programs, and automate aspects of probabilistic inference (Goodman et al., 2014). Some PPLs support LMs as parts of models (Dohan et al., 2022; Lew et al., 2020; 2023), and LMs have also been used to generate probabilistic programs over symbolic domains (Li et al., 2024; Wong et al., 2023). More recently, PPLs have evolved to feature *programmable inference* (Mansinghka et al., 2014; 2018), so that programs can concisely specify both models and custom inference algorithms. We use LMs to generate code in LLAMPPL (Lew et al., 2023), a PPL built on a modern LM stack that enables users to define inference procedures via short, intuitive Python programs.

## 3 DISCIPL

### 3.1 General Framework

A *language model M* is a distribution on token sequences from a vocabulary $\boldsymbol{x} \in \mathcal{V}^*$ whose probability factorizes autoregressively as $p_M(\boldsymbol{x}) = \prod_{t=1}^{|x|} p_M(x_t \mid \boldsymbol{x}_{<t})$. We designate roles for two LMs, a **Planner** $M_P$ and a **Follower** $M_F$. We assume that both support efficient *conditional generation*, and that $M_F$ also supports efficient *conditional probability evaluation*.

---

**Algorithm 1** DISCIPL Outer Loop

---

1: **function** DISCIPL(*task* $d_{\text{task}}$, *Planner* $M_P$, *Follower* $M_F$, *budget* $N$, *retries* $R$, *engine* $\mathcal{I}$)
2:     $result \leftarrow \varepsilon_{\text{null}}, r \leftarrow 1$         ▷ *Initialize with empty string*
3:     **while** *result* $\in \mathcal{E}$ **and** $r \leq R$ **do**
4:        $\pi \sim p_{M_P}(\cdot \mid \mathtt{prompt}(d_{\text{task}}, result))$    ▷ *Generate inference program (with prior errors as feedback)*
5:        $result \sim \mathcal{I}(\pi, M_F, N)$        ▷ *Execute the program with Follower*
6:        **if** *result* $\in \mathcal{E}$ **then**
7:           $r \leftarrow r + 1$        ▷ *Retry on runtime errors*
     **return** *result*

---

The key idea in DISCIPL is that $M_P$ can generate an *inference program* $\pi$ in some language $\mathcal{L}$ that describes how $M_F$ should be used to solve a task. The program may make multiple asynchronous queries to $M_F$, in both generative (i.e., sampling) and evaluative (i.e., probability computation) modes. The language provides an *inference engine* $\mathcal{I}$ that takes a program $\pi \in \mathcal{L}$, a Follower model $M_F$, and an *inference budget* $N \in \mathbb{N}$, and yields samples from a distribution on results, which can either be *answers* $x \in \mathcal{V}^*$ or *errors* $\varepsilon \in \mathcal{E}$.

With these ingredients, the basic DISCIPL algorithm (Alg. 1) proceeds as follows. Given a natural language task description $d_{\text{task}}$, we first prompt the Planner $M_P$ to generate a program in $\pi \in \mathcal{L}$ suitable for solving the task. We then attempt to run $\pi$ using the inference engine $\mathcal{I}$. If inference succeeds, we return its output. Otherwise, we re-prompt $M_P$ to correct $\pi$, passing the error $\varepsilon$ and associated traceback, up to a max number of retries $R$.

### 3.2 Our Instantiation: Language Model Probabilistic Programs

**Language of inference programs.** In our instantiation of DISCIPL, programs $\pi$ are written in LLAMPPL (Lew et al., 2023), a Python framework for probabilistic programming with language models. LLAMPPL programs work by repeatedly *extending* candidate generations by one or more tokens, and then *scoring* the proposed extensions. The programmer (in our case, the Planner model) can customize the process by which new extensions are proposed and scored; behind the scenes, LLAMPPL's inference engine automatically coordinates the overall search, maintaining multiple candidate generations in parallel, and dynamically reallocating computational resources to high-scoring partial completions.

As a concrete example, consider the program in Fig. 1, whose `step` method stochastically adds one word to the running generation, either sampling from $M_F$ (using `sample`) or forcing it to agree with a word constraint (using `observe`). The `observe` method has the side-effect of updating the candidate's *score*, multiplying it by the probability the Follower assigns to the observed word. The illustration to the right of the program shows how these scores are used by the inference engine: when the word "Glasgow" is forced after the prefix "The year begins," the candidate generation's score drops sharply (due to the low probability of "Glasgow"), and the generation is culled.

More generally, programs can use arbitrary logic to extend a partial completion $s$ and to multiply an *update w* into that completion's running score. These score updates can encode hard constraints (0 indicates violation), or soft constraints, based on symbolic heuristics, LM-as-judge schemes, or token probabilities under the Follower. They can also incorporate *importance weights* that correct for biases in the proposal process (see §3.3).

**Mathematical interpretation of inference programs.** An inference program $\pi$ provides a blueprint for sequential *inference algorithms* that search for high-quality completions. Formally, we say that the goal of these algorithms is to generate samples from a *target distribution* induced by the program that assigns high probability to high-scoring sequences. Let $\sigma : \mathcal{V}^* \to P(\mathcal{V}^* \times \mathbb{R})$ denote a step function that maps a starting string $s$ to a distribution $\sigma(s)$ over updated strings $s' \in \mathcal{V}^*$ and multiplicative score updates $w \in \mathbb{R}$. Then for a fixed maximum generation length $T \in \mathbb{N}$, the target distribution is defined as

$$p_{target}(s) \propto \mathbb{E}_{(s'_1, w_1) \sim \sigma(\varepsilon_{\text{null}}), \dots, (s'_T, w_T) \sim \sigma(s'_{T-1})} \left[ \mathbb{1}[s'_T = s] \cdot \prod_{i=1}^{T} w_i \right]. \tag{1}$$

**(A) Token masking**

```
# Generates exactly 10 chars
text = "", K = 10
while len(text) <= K:
  # Sample from mask and
  # auto-update weight
  async with TokenLengthMask(
    max_chars=K-len(text)
  ):
    text += \
      await self.next_token()
```

**(B) Self-hinting**

```
# Generates gift list under a budget
async def step(self):
  # Update Follower LM prompt
  await self.hint(
    f"You have ${self.budget} left!"
  )
  gift = await self.extend(
    start="-", stop="\n"
  )
  self.budget -= self.extract_price(gift)
  self.condition(self.budget >= 0)
```

**(C) Self-checking**

```
# Checks if text is a valid Haiku
def check(self, text: str) -> bool:
  lines = text.strip().split()
  if len(lines) != 3:
    return False
  # Check syllable structure
  for l, s in zip(lines, [5, 7, 5]):
    if textstat.syllable_count(l) != s:
      return False
  return True
```

Figure 2: **Inference programming patterns for self-steering**, discussed in §3.3.

Intuitively, this equation defines the target probability of string $s$ to be proportional to the probability of generating $s$ via repeated application of step, upweighted or downweighted according to the accumulated score.

**Scalable test-time search via probabilistic inference.** Our inference engine $\mathcal{I}$ implements several general-purpose Monte Carlo methods for approximately sampling from the target distribution $p_{target}$. All methods support *parallel scaling* based on an inference budget $N$. Our experiments (§4) evaluate three instantiations of DISCIPL.

*Importance Sampling (IS).* Importance sampling generates $N$ full completions $s_1, \ldots, s_N$ in parallel, by initializing $N$ empty generations and repeatedly calling step until all have completed. For each candidate $s_i$, the score updates $w$ from each step are multiplied to obtain an overall score $w^{(i)}$. Lastly, a final output is sampled $s^*$ from the distribution $p_{IS}(s^{(i)}) = w^{(i)}/\sum_{j=1}^{N} w^{(j)}$. When $p_{target}$ is well-defined, $p_{IS}$ converges to $p_{target}$ as $N \to \infty$.

*Sequential Monte Carlo (SMC).* Like IS, SMC (Doucet et al., 2001, and see Alg. 2) initializes $N$ empty generations, called *particles*, with associated weights initialized to 1. SMC alternates between (1) calling step on all particles in parallel and multiplying the returned score updates into the particle weights; and (2) *resampling* to cull low-scoring particles and replace them with copies of high-scoring particles. (All weights are reset to uniform after resampling.) Maintaining multiple copies of promising particles allows each to be extended independently by the next call to step. This has the effect of adaptively reallocating the computation budget ($N$) to focus on promising candidates. As in IS, the final particle $s^*$ is sampled with probability proportional to the weights. Under mild technical conditions, the SMC sampling distribution also converges to $p_{target}$ as $N \to \infty$.

*Rejection Sampling (RS).* When a program's step method generates an entire completion and then computes a binary score update, running $\mathcal{I}$ (with budget $N$) reduces to rejection sampling (generating $N$ samples and checking whether any satisfy the constraint). We implement this pattern as DISCIPL-RS in §4.

### 3.3 Common Patterns

Programs in DISCIPL adhere to several common inference patterns, for which we have implemented library support and which are also illustrated in the prompt to the Planner.

**Step-by-step problem decomposition.** The Planner must decide how to decompose a task into a sequence of extend-and-score steps; this determines how often different candidates are compared and resampled. A common pattern is to make each step extend by a task-relevant unit (e.g., a line of a poem, a word of a sentence with word-level constraints, etc.).

**Prior and proposal prompts.** Imposing constraints can lead LMs to produce incoherent generations. For example, when prompted to generate a sentence using the words "dog," "throw," and "frisbee," small LMs yield semantically dubious completions like, "Two dogs are throwing frisbees at each other" (Lin et al., 2020). To promote coherency, programs can compensate for biases in the *proposal* distribution, which is aware of task-specific constraints, with scores from a *prior*, which ensures fluency. The Planner defines the prior and proposal distributions via separate prompts. Typically, the prior prompt defines more general instructions, e.g., "Write a sentence that is grammatically correct and makes sense."

**Constrained generation with weight correction (Fig. 2A).** In many situations, we might want the Follower to generate specific token sequences (e.g., "Glasgow"), or more generally, to adhere to formal constraints like regular expressions or grammars. The Planner can apply *token masks* that both enforce these constraints at generation time, and automatically incorporate *importance weights* that correct for the distortion in the LM's distribution resulting from the mask (Loula et al., 2025; Park et al., 2024).

**Self-hinting (Fig. 2B).** Since the Planner controls the Follower's proposal prompt, one powerful pattern is to dynamically update it to reflect stateful information relevant to the next generation step. We expose a special `hint()` method that injects "`Note to self: {hint}`" into the Follower's context, where the hint can include text as well as Python variables and objects. This technique functions as a generalized calculator (Cobbe et al., 2021) that can perform arbitrary symbolic computations and pass their results to the Follower.

**Self-checking (Fig. 2C).** While programs often ensure correctness by construction, some problems cannot be verified until generation is complete. In other cases, it may still be preferable to use guess-and-check over constrained generation, or to catch bugs in the inference logic. For this reason, the Planner defines a distinguished `check()` method, which (like everything it generates) can make use of external libraries.

## 4 Experiments

### 4.1 Domains

**Constrained generation.** We evaluate DISCIPL on COLLIE-v1 (Yao et al., 2024), a constrained generation benchmark designed as a challenge dataset for LMs. Tasks in COLLIE (Table 2) are composed from a formal grammar of constraints specified at multiple levels of text granularity; here, we focus on the sentence and paragraph levels. The combinatorial nature of the grammar is what makes COLLIE tasks challenging—Yao et al. (2024) find that overall performance ranges from 16.3% (Alpaca-7B) to 50.9% (GPT-4).

**Puzzles.** To evaluate generalization beyond tasks that can be defined with a grammar, we construct a mini-dataset of challenging naturalistic generation tasks. PUZZLES (Table 5) consists of four task types that require models to compose structured poetry, write a grant proposal abstract subject to various constraints, generate an ingredients list that meets a monetary budget, and plan a multi-day travel itinerary.

### 4.2 Evaluation Metrics

**Validity.** We are interested in building systems that effectively leverage test-time compute to improve their answers to hard-to-answer queries. Accordingly, we focus on expected Pass@1, which (unlike Pass@k) models a setting that does not assume access to an oracle verifier. We implement a generalized version of the standard unbiased Pass@k estimator (Brown et al., 2024; Chen et al., 2021) that uses the weights computed during inference (if available) and uniform weights for baselines (App. D).

**Coherency.** As highlighted in the example in Fig. 1, the introduction of constraints can impact the fluency of generated text. To assess coherency, we adopt the LLM-as-judge evaluation from Yao et al. (2024) using GPT-4o-mini with a zero-shot prompt (App. E). While coherency could also be assessed with human ratings, this approach offers a scalable, affordable, and reasonably reliable measure of the overall text quality.

### 4.3 Experiment Setup

Our experiments evaluate whether DISCIPL can generate effective and efficient inference programs for solving unseen tasks in a stratified cross-validation setup. We begin by manually writing an example inference model for a single instance of each task type in COLLIE and PUZZLES. We few-shot prompt the Planner LM with the examples from the corresponding domain (App. K.2), holding out the model corresponding to the target task type. (For PUZZLES, we also include the examples from COLLIE.)

**Planner and Follower LMs.** The goal of the Planner is to generate an `InferenceModel` subclass that implements various methods, including `step()` and `check()`. We use a system prompt in the style of a `README.md` (App. K.1, which also serves as a condensed tutorial on LLAMPPL for the interested reader) to instruct to the Planner how to write inference models. We instantiate the Planner with a capable LM (`gpt-4o-2024-08-06`; OpenAI, 2024a) that can attend to detailed instructions; however, the Planner does not itself perform any reasoning outside generating Python code. Except where otherwise indicated in §5.3, we instantiate the Follower with `Llama-3.2-1B-Instruct` (Grattafiori et al., 2024).

**Baselines.** We benchmark DISCIPL against several baselines:

- **Follower-only:** We prompt $M_F$ to directly solve the task, sampling $N$ independent completions with `temperature=1.0`. We also run a variant with stochastic beam search where $N$ controls the beam size.
- **Planner-only:** We prompt $M_P$ to solve the task both directly and with chain-of-thought (CoT; App. F.1). For comparison, we also benchmark a smaller model (`gpt-4o-mini-2024-07-18`) in the same family.
- **Reasoning model:** As a representative frontier CoT reasoning model, we evaluate o1 (`o1-2024-12-17`; OpenAI, 2024b).[1] We note that o1 is both theoretically and empirically more expensive than DISCIPL (see App. J for a detailed analysis of compute costs). Accordingly, o1 is primarily intended as a skyline on task performance, and is not directly comparable in terms of resource efficiency.

**Expert programs.** We evaluate a variant where we grant the Planner access to the reference implementation corresponding to the target task. This oracle condition (denoted DISCIPL*) evaluates the extent to which the Planner is able to recover the expert programs.

## 5 Results and Analysis

We report the main results of our experiments in Table 1 with figures breaking down sentence (Fig. 3), paragraph (Fig. 8), and PUZZLES (Fig. 9) performance by task.

### 5.1 Validity

**Follower-only baseline is unreliable.** Perhaps unsurprisingly given its size, Llama-1B is only weakly able to follow the task instructions, scoring poorly on COLLIE sentence tasks (Pass@1=0.07) and PUZZLES (Pass@1=0.08) and moderately on COLLIE paragraph (Pass@1=0.38) tasks. Notably, off-the-shelf decoding methods (e.g., beam search), which optimize for sequence log probability, do not improve performance on the constrained generation objective.

**Planner-only baseline performance is task-dependent.** While GPT-4o and GPT-4o-mini are nearly perfect at including specific words (e.g., sent_04), they struggle to position keywords at locations other than the start of a sentence (e.g., sent_02, para_05) and cannot reliably count characters (e.g., sent_01, sent_03). CoT prompting yields small improvements on COLLIE sentences and PUZZLES; however, the reasoning is still incorrect in many cases (e.g., App. F.2). These results are in line with Yao et al. (2024), who found that GPT-4 performed poorly on these tasks.

**Reasoning model shows strong (though not perfect) performance.** With the benefit of ample test-time compute, o1 achieves near-ceiling Pass@1 on COLLIE tasks. Nevertheless, Pass@1 for o1 dips below 1.0 on several tasks (sent_01, para_03, para_05, and `ingredients_list`).

**DISCIPL significantly boosts performance across domains.** On all tasks, DISCIPL-SMC and DISCIPL-IS far exceed the Follower-only baselines, enabling various skills like character counting and word positioning that are entirely absent from Llama-3.2-1B. On paragraph tasks, DISCIPL closes most of the gap between the Follower and Planner performance, bringing Llama up to GPT-4o/GPT-4o-mini level (Fig. 8). Meanwhile, on sentence

---

[1]Following OpenAI's guidelines, we query o1 with `max_tokens=25000`. Due to cost, we only query o1 once per task instance.

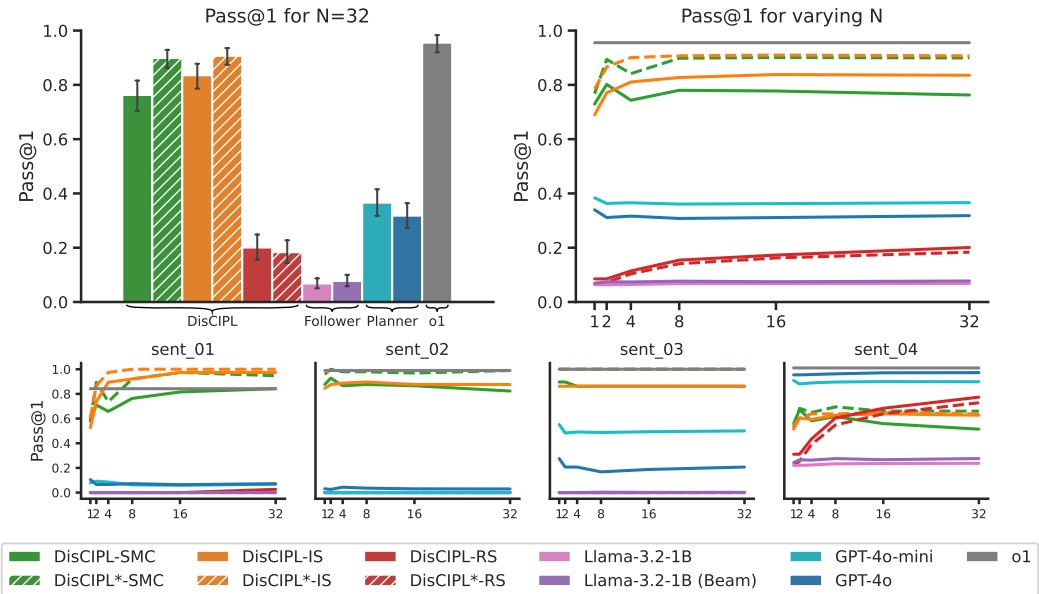

Figure 3: **Validity on COLLIE Sentence-Level Tasks.** (*Top left*) Average pass rate (Pass@1) with a fixed inference budget (N=32) corresponding to the number of samples or particles. (*Top right*) Pass@1 under a varying sample budget. While pass rates vary by task (*bottom row*), overall DISCIPL surpasses *both* the Follower and Planner baselines and approaches the performance a strong reasoning model (o1). Moreover, the performance of autogenerated inference programs nearly matches that of expert programs (DISCIPL*).

| Method | Sampling Method | Model | COLLIE Sentence | | COLLIE Paragraph | | PUZZLES | |
| | | | Pass@1 | Coherency | Pass@1 | Coherency | Pass@1 | Coherency |
| --- | --- | --- | --- | --- | --- | --- | --- | --- |
| DisCIPL | SMC | Llama-3.2-1B | 0.76 | 5.61 | 0.69 | 6.72 | 0.42 | 6.40 |
| | Importance (IS) | Llama-3.2-1B | 0.84 | 5.07 | 0.66 | 5.89 | 0.38 | 5.65 |
| | Rejection (RS) | Llama-3.2-1B | 0.20 | 8.22 | 0.62 | 8.48 | 0.25 | 8.85 |
| Follower-only | Standard | Llama-3.2-1B | 0.07 | 8.12 | 0.38 | 8.28 | 0.08 | 8.60 |
| | Beam Search | Llama-3.2-1B | 0.08 | 8.93 | 0.38 | 8.96 | 0.03 | 8.68 |
| Planner-only | Standard | GPT-4o-mini | 0.37 | 9.00 | 0.72 | 9.10 | 0.27 | 9.20 |
| | | GPT-4o | 0.32 | 8.79 | 0.80 | 9.16 | 0.25 | 9.43 |
| | | GPT-4o (CoT) | 0.46 | 8.73 | 0.76 | 8.79 | 0.30 | 9.28 |
| Reasoning | Standard | o1 | 0.96 | 7.69 | 0.95 | 8.53 | 0.82 | 9.00 |

Table 1: **Summary of results from all experiments.** Pass@1 measures expected validity. Coherency (10-point scale) measures overall fluency for all generations (including invalid ones).

tasks, DISCIPL surpasses both Follower and Planner baselines, and approaches o1-level performance (Fig. 3). Finally, on PUZZLES, DISCIPL on average outperforms both Planner and Follower baselines, though performance varies between tasks and there is a bigger gap between autogenerated and expert programs.

**DISCIPL knows when it is wrong.** Across all tasks, autogenerated check() methods closely match ground truth verifiers (Fig. 5). However, rejection sampling (RS) produces substantially fewer valid generations than combining step() and check() (SMC and IS).

## 5.2 Coherency

What inference algorithms can we leverage to push the Pareto frontier of coherency/validity? Our results suggest that SMC (Fig. 4) is well-suited to this goal—in our experiments, SMC consistently achieves higher coherency scores than IS for comparable Pass@1. One way of understanding this result is through Fig. 1, which illustrates how SMC resampling filters out particles that satisfy constraints but introduce disfluencies. This also helps to explain why, on some tasks, the Pass@1 curves for DISCIPL-SMC appear relatively flat: in

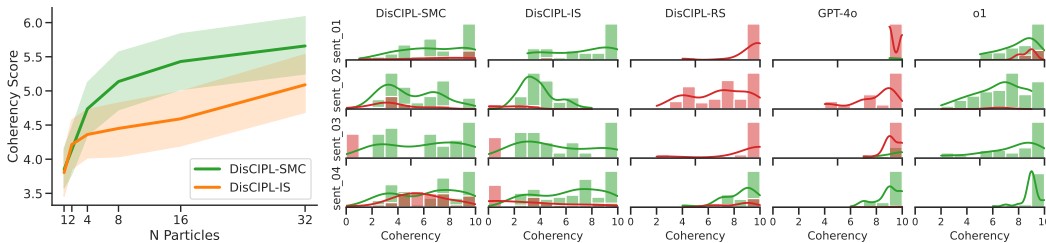

Figure 4: **Coherency.** (*Left*) SMC resampling leads to more coherent generations and scales with compute budget. (*Right*) Distributions of coherency scores on COLLIE sentence tasks (densities are normalized per-subplot). While non-reasoning LMs produce coherent but invalid text (red), DISCIPL enables Llama to satisfy constraints (green) with improving coherency as the particle count scales.

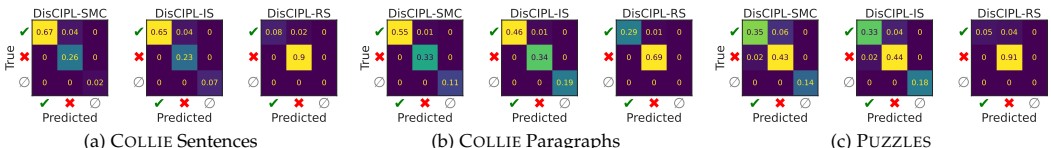

Figure 5: **Classification accuracy of autogenerated check() methods.** Across domains, DISCIPL aligns closely with the ground truth verifiers. ($\varnothing$ denotes null results due to runtime errors; App. G.)

these cases, the inference programs ensure validity *by construction*, so the benefits of scaling show up instead in the coherency scores.

While using a separate proposal and prior (§3.3) generally improves coherency, in some cases, it may backfire: for instance, on sent_04 we see that Pass@1 *decreases* with N > 8 (Fig. 3). For this task, the Planner LM showed a strong inductive bias towards a word-level step, even when provided expert programs. However, this strategy will *filter out* particles containing target words, which have low probability under the prior. A smarter Planner could easily avoid this issue by choosing a step granularity that better aligns with the constraints (e.g., sampling multiple words until a target is generated).

### 5.3 Scaling the Follower Model

While our main experiments instantiate DISCIPL with a specific Follower LM (Llama-3.2-1B), in other settings, we may wish to utilize more capable Follower models. Here, we explore the effects of two different aspects of model choice—size and architecture—using COLLIE as a controlled setting to characterize performance tradeoffs.

**Model Size (Fig. 6, top)** First, we hold the architecture constant and scale up the size of the Follower from 1B $\rightarrow$ 3B $\rightarrow$ 8B within the Llama-3 family.[2] As expected, the baselines improve with model size; nevertheless, for sentence tasks, these still significantly underperform DISCIPL-SMC/IS. Moreover, we find that baseline Llama-3 coherency scales *inversely* with Pass@1; intuitively, the larger Llama models appear to try harder to adhere to the constraints, which results in less natural text. In contrast, under DISCIPL-SMC/IS, Pass@1 and coherency improve simultaneously with Follower size (see App. B for full numbers). These results illustrate that even for a fixed Planner LM, our inference programs nevertheless benefit from being executed by more capable Followers.

**Model Family (Fig. 6, bottom)** We instantiate DISCIPL with Qwen3 (Yang et al., 2025), a newer LM that outperforms Llama-3 on various benchmarks and also provides lightweight models of comparable size. On sentence tasks, we find that Qwen3-1.7B moderately outperforms Llama-3.2-1B at baseline, and that these benefits translate to DISCIPL (+0.07 Pass@1). (On paragraph tasks, we do not find a significant difference between models.) As these results highlight, DISCIPL reflects the capabilities of the underlying Follower LM, and stands to benefit as advances in distillation yield increasingly capable small LMs.

---

[2]Since the largest text-only model in the Llama-3.2 family is 3B parameters, we use Llama-3.1 for the 8B model. This reversion may explain apparent inconsistencies in scaling trends, such as the drop from 3B $\rightarrow$ 8B for DISCIPL-RS seen in Fig. 6(a).

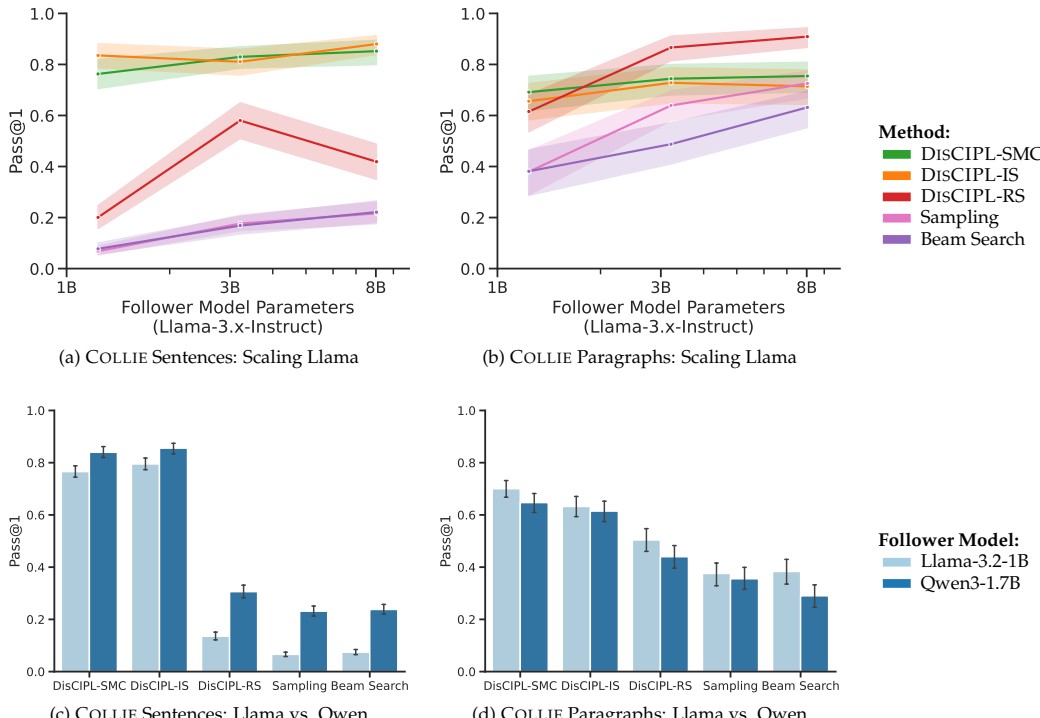

Figure 6: **Characterizing effects of Follower model size and architecture.** (*Top row*) Scaling size for a fixed model family (Llama-3 1B, 3B, and 8B). (*Bottom row*) Comparing two LMs with similar size (Llama-3 vs. Qwen3).

## 6    Limitations, Future Work, and Conclusions

The results presented here represent an initial validation of the broad framework outlined in §3. In this section, we acknowledge several limitations of the current instantiation of DISCIPL and highlight several promising directions for follow-up work.

**Generalization.** In addition to other constrained generation settings (e.g., Lin et al., 2020; Zhou et al., 2023a), self-steering can also be extended to mathematical reasoning (Lightman et al., 2024; Wang et al., 2024a) as well as domains with soft constraints (e.g., steering based on reward models).

**Inference Algorithms.** While many text generation tasks are amenable to sequential inference, some problems may be more efficiently solved with backtracking (e.g., MCTS; Coulom, 2007; Kocsis & Szepesvári, 2006) or iterative editing (Welleck et al., 2023), both of which are implementable as extensions to DISCIPL.

**Self-Improvement.** Since generating inference programs requires non-trivial reasoning, we instantiate the Planner with a larger and more capable LM than the Follower. However, in principle, we could use the same LM to play these two roles. This recursive "self-steering" setup could learn by bootstrapping (e.g., Zelikman et al., 2022) or library learning (Ellis et al., 2021; Grand et al., 2024).

In conclusion, in this work, we introduced DISCIPL, a general framework for problem-solving with language models that orchestrates inference-time compute by writing and executing inference programs. We believe our approach offers a unifying probabilistic perspective on inference-time computation with LMs, as well as a practical tools for automating inference engineering. Our results demonstrate the potential of self-steering to enable accurate and efficient parallel inference with populations of small LMs, with performance rivaling much larger and more costly frontier models. In future work, we aim to generalize self-steering language models to new problem domains and inference patterns.

*Acknowledgments*

We would like to thank Timothy O'Donnell, João Loula, Ced Zhang, Cédric Colas, Noah Goodman, Aniruddha Nrusimha, Linlu Qiu, Tan Zhi-Xuan, and Lionel Wong for helpful discussions and feedback. Special thanks to Ben LeBrun and the GenLM team for engineering support.

The authors gratefully acknowledge support from the MIT Quest for Intelligence, the MIT-IBM Watson AI Lab, the Intel Corporation, AFOSR, DARPA, ONR, and the National Science Foundation (NSF) under grants CCF-2217064 and IIS-2238240. G.G. is supported by a NSF Graduate Research Fellowship under Grant No. 2141064. J.B.T. is supported by AFOSR, the MIT Quest for Intelligence, the MIT-IBM Watson AI Lab, ONR Science of AI, and Siegel Family Endowment. V.K.M. and A.K.L. are supported by an anonymous philanthropic gift as well as gifts from Siegel Family Foundation that support the MIT Quest for Intelligence. J.A. is supported by a Sloan Research Fellowship. Any opinions, findings, and conclusions or recommendations expressed in this material are those of the author(s) and do not necessarily reflect the views of sponsors.

*Author Contributions*

**Gabriel Grand (primary author)**: Research conception, narrative development, software design and implementation, experiments, analysis of results, figure-making, writing.

**Joshua B. Tenenbaum:** Senior mentorship, narrative development.

**Vikash K. Mansinghka:** Senior mentorship, narrative development.

**Alexander K. Lew:** Senior mentorship, research conception, narrative development, mathematical formalisms, analysis of results, figure-making, writing.

**Jacob Andreas:** Senior mentorship, research conception, narrative development, analysis of results, writing.

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

# Appendix

## Table of Contents

# A    SMC Visualization

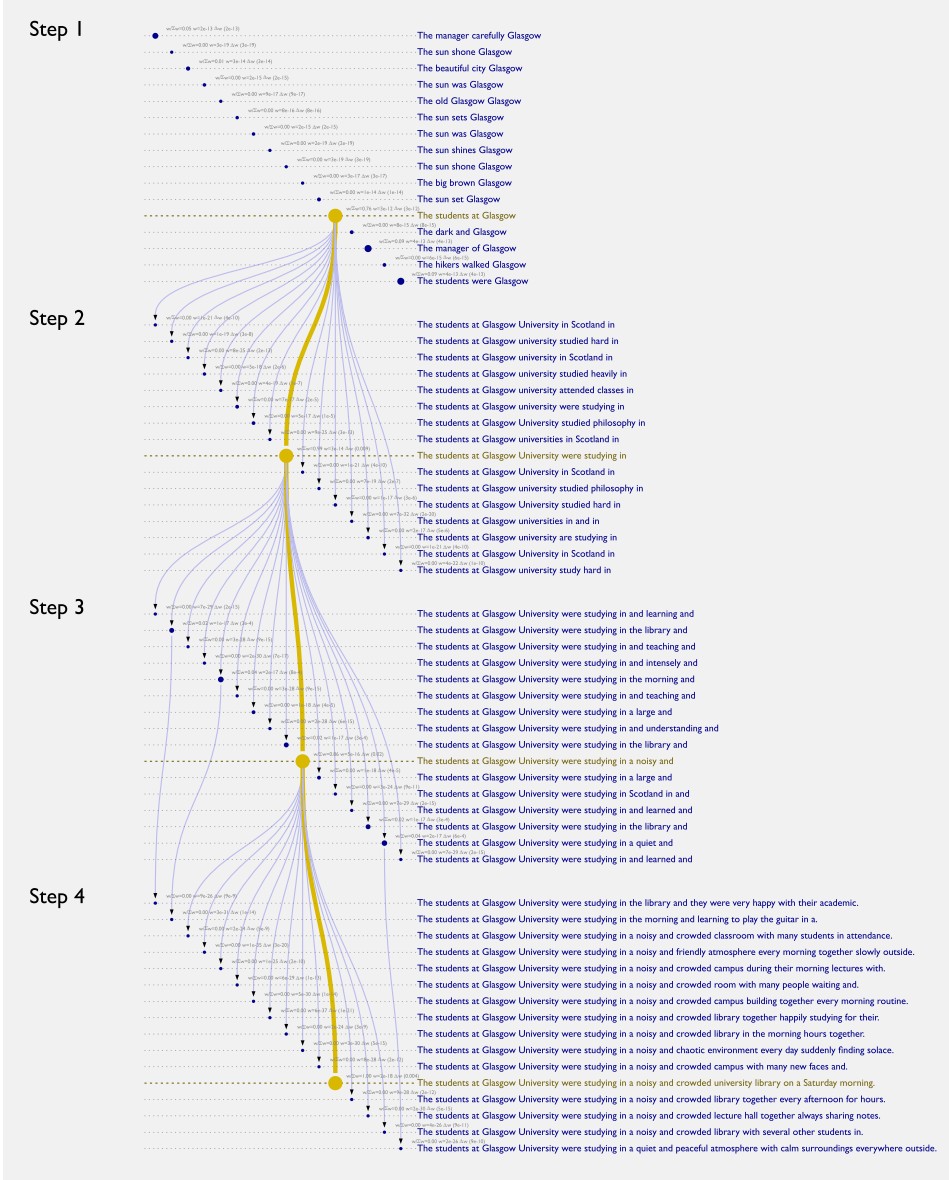

Figure 7: **Inference in action.** Visualization of SMC inference for a DISCIPL program for the COLLIE sent_02 task (Fig. 1*): "Please generate a sentence: 1) with exactly 18 words; 2) with the 4th, 8th, 11th words to be 'Glasgow', 'in', 'and' respectively." Weights for N=16 particles are computed after each step and correspond intuitively to a measure of fluency under the constraints. For instance, after Step 1, particles for which "Glasgow" is a natural 4th word are propagated (e.g., "The students at Glasgow"), while others are filtered out (e.g., "The big brown Glasgow"). Each step corresponds to a single constraint: the first three steps each generate until the next target word, and the final step generates until the 18th word. In this way, the inference program ensures validity by construction, while adaptive resampling via SMC selects for overall coherency.

*Note that for pedagogical reasons, some details have been simplified in the program that appears in Fig. 1. In particular, in Fig. 1, each step generates one word; however, in the program that produced this visualization, each step generates multiple words until the next constraint has been satisfied. Both approaches are valid, but the multi-word approach is more efficient and leads to better coherency, as discussed in §5.2.*

## B  COLLIE Dataset and Results

We evaluate on a subset of 9/13 of the tasks in COLLIE-v1 (Yao et al., 2024) corresponding to the sentence and paragraph levels.[3] We use the task instances drawn from the Wikipedia corpus of COLLIE-v1. Since the number of instances varies significantly across task types, when computing aggregate metrics, we normalize with respect to task-level (sentence and paragraph), such that each task instance is treated as having been sampled from a uniform prior over constraint types.

| TASK | N | EXAMPLE PROMPT |
|---|---|---|
| sent_01 | 38 | Please generate a sentence with exactly 82 characters. Include whitespace into your character count. |
| sent_02 | 98 | Please generate a sentence: 1) with exactly 11 words; 2) with the 4th, 8th, 11th words to be 'Series', 'and', '4' respectively. |
| sent_03 | 29 | Please generate a sentence: 1) with at least 9 words; 2) with all words having at most 7 characters. |
| sent_04 | 94 | Please generate a sentence containing the word 'have', 'rising', 'the'. |
| para_01 | 9 | Please generate a paragraph with all sentences having the 1st word to be 'The'. |
| para_02 | 94 | Please generate a paragraph: 1) with exactly 3 sentences; 2) not containing the word 'be'; 3) not containing the word 'this'; 4) not containing the word 'is'. |
| para_03 | 93 | Please generate a paragraph: 1) with exactly 4 sentences; 2) with all sentences having at least 12 words; 3) with all sentences having at most 20 words. |
| para_04 | 18 | Please generate a paragraph: 1) with at least 3 sentences; 2) with all sentences having at least 21 words. |
| para_05 | 89 | Please generate a paragraph: 1) with exactly 3 sentences; 2) with sentences having the last word to be 'convention', 'president', 'Wisconsin' respectively. |

Table 2: **Summary of tasks in COLLIE-v1 used for our evaluation.**

| Method | Sampling | Model | sent_01 Pass@1 | sent_01 Coh. | sent_02 Pass@1 | sent_02 Coh. | sent_03 Pass@1 | sent_03 Coh. | sent_04 Pass@1 | sent_04 Coh. | Overall Pass@1 | Overall Coh. |
|---|---|---|---|---|---|---|---|---|---|---|---|---|
| DisCIPL | SMC | Llama-3.2-1B | 0.84 | 6.61 | 0.81 | 4.80 | 0.86 | 5.07 | 0.53 | 5.96 | 0.76 | 5.61 |
| | | Llama-3.2-3B | 0.97 | 7.71 | 0.84 | 4.34 | 0.97 | 7.28 | 0.54 | 6.12 | 0.83 | 6.36 |
| | | Llama-3.1-8B | 0.89 | 6.79 | 0.87 | 5.34 | 0.97 | 5.14 | 0.68 | 7.03 | 0.85 | 6.07 |
| | | Qwen3-1.7B | 0.76 | 6.21 | 0.92 | 2.31 | 0.90 | 7.24 | 0.73 | 5.05 | 0.83 | 5.20 |
| DisCIPL | Importance | Llama-3.2-1B | 0.97 | 6.84 | 0.87 | 3.55 | 0.86 | 4.55 | 0.64 | 5.34 | 0.84 | 5.07 |
| | | Llama-3.2-3B | 1.00 | 7.82 | 0.83 | 4.04 | 0.86 | 5.55 | 0.55 | 5.49 | 0.81 | 5.72 |
| | | Llama-3.1-8B | 0.97 | 7.71 | 0.86 | 4.73 | 0.97 | 4.31 | 0.72 | 6.59 | 0.88 | 5.84 |
| | | Qwen3-1.7B | 0.95 | 5.21 | 0.92 | 1.86 | 0.86 | 6.86 | 0.81 | 5.13 | 0.88 | 4.76 |
| DisCIPL | Rejection | Llama-3.2-1B | 0.03 | 9.05 | 0.00 | 6.14 | 0.00 | 9.41 | 0.78 | 8.28 | 0.20 | 8.22 |
| | | Llama-3.2-3B | 0.61 | 7.92 | 0.00 | 5.28 | 0.90 | 7.28 | 0.82 | 8.67 | 0.58 | 7.29 |
| | | Llama-3.1-8B | 0.39 | 8.26 | 0.01 | 5.26 | 0.34 | 5.90 | 0.93 | 8.71 | 0.42 | 7.03 |
| | | Qwen3-1.7B | 0.05 | 8.21 | 0.01 | 4.47 | 0.55 | 8.59 | 0.90 | 8.02 | 0.38 | 7.32 |
| DisCIPL* expert programs | SMC | Llama-3.2-1B | 0.95 | 7.29 | 0.98 | 5.06 | 1.00 | 7.76 | 0.67 | 6.32 | 0.90 | 6.61 |
| | Importance | Llama-3.2-1B | 1.00 | 6.61 | 0.98 | 4.06 | 1.00 | 6.03 | 0.65 | 6.21 | 0.91 | 5.73 |
| | Rejection | Llama-3.2-1B | 0.00 | 9.55 | 0.00 | 6.53 | 0.00 | 9.90 | 0.73 | 8.21 | 0.18 | 8.55 |
| Follower-only | Standard | Llama-3.2-1B | 0.00 | 9.24 | 0.00 | 6.01 | 0.00 | 8.86 | 0.27 | 8.36 | 0.07 | 8.12 |
| | | Llama-3.2-3B | 0.02 | 8.37 | 0.00 | 5.51 | 0.12 | 7.76 | 0.56 | 8.11 | 0.18 | 7.44 |
| | | Llama-3.1-8B | 0.02 | 7.95 | 0.00 | 5.23 | 0.02 | 5.76 | 0.82 | 8.20 | 0.22 | 6.79 |
| | Beam Search | Llama-3.2-1B | 0.00 | 9.61 | 0.00 | 7.90 | 0.00 | 9.76 | 0.31 | 8.47 | 0.08 | 8.93 |
| | | Llama-3.2-3B | 0.03 | 8.76 | 0.00 | 6.39 | 0.05 | 9.24 | 0.60 | 8.68 | 0.17 | 8.27 |
| | | Llama-3.1-8B | 0.02 | 8.37 | 0.00 | 5.76 | 0.01 | 8.79 | 0.85 | 8.69 | 0.22 | 7.90 |
| | Standard | Qwen3-1.7B | 0.01 | 8.29 | 0.00 | 4.55 | 0.08 | 8.79 | 0.82 | 7.70 | 0.23 | 7.33 |
| | Beam Search | Qwen3-1.7B | 0.01 | 5.95 | 0.00 | 3.96 | 0.15 | 7.90 | 0.80 | 7.69 | 0.24 | 6.37 |
| Planner-only | Standard | GPT-4o-mini | 0.07 | 9.63 | 0.00 | 8.10 | 0.50 | 9.55 | 0.89 | 8.71 | 0.37 | 9.00 |
| | | GPT-4o | 0.07 | 9.21 | 0.03 | 7.78 | 0.21 | 9.21 | 0.96 | 8.97 | 0.32 | 8.79 |
| | | GPT-4o (CoT) | 0.07 | 8.87 | 0.08 | 7.89 | 0.71 | 9.31 | 0.98 | 8.84 | 0.46 | 8.73 |
| Reasoning | Standard | o1 | 0.84 | 8.03 | 0.98 | 6.48 | 1.00 | 7.34 | 1.00 | 8.90 | 0.96 | 7.69 |

Table 3: **COLLIE Sentence-Level Results.**

---

[3] While COLLIE also defines several word-level constraints, we found that, even with tools for fine-grained token masking, these present a particular challenge for LMs due to the problem of token misalignment. While it is possible to recover next-character distributions from token-level LMs (e.g., Vieira et al., 2024), such approximations are expensive to compute. Instead, we focus on constraints at the sentence-level and higher that can be effectively expressed with token-level LMs.

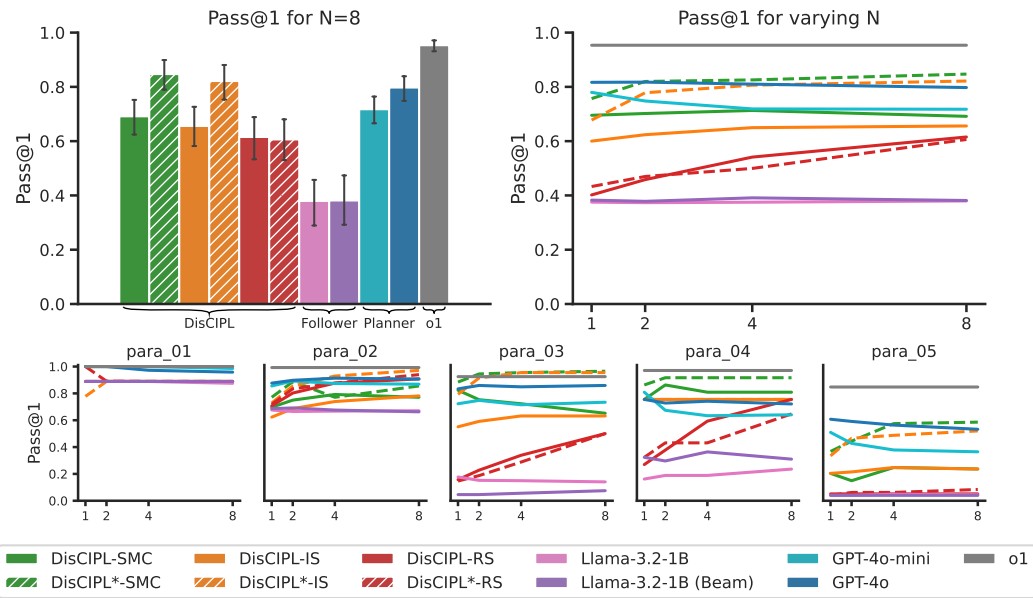

Figure 8: **Validity on COLLIE Paragraph-Level Tasks.** Figure structure is identical to COLLIE sentence-level tasks (Fig. 3). Since generations are longer, the total sampling budget is limited to $N = 8$. Baseline performance is higher for all models as these tasks appear to be more in-distribution for LMs. Nevertheless, we observe that DISCIPL helps to close the Pass@1 gap between the Follower-only (Llama-3.2-1B) and Planner-only (GPT-4o) baselines.

| Method | Sampling | Model | para_01 Pass@1 | Coh. | para_02 Pass@1 | Coh. | para_03 Pass@1 | Coh. | para_04 Pass@1 | Coh. | para_05 Pass@1 | Coh. | Overall Pass@1 | Coh. |
|---|---|---|---|---|---|---|---|---|---|---|---|---|---|---|
| DisCIPL | SMC | Llama-3.2-1B | 1.00 | 7.00 | 0.78 | 8.02 | 0.65 | 5.92 | 0.83 | 8.94 | 0.20 | 3.70 | 0.69 | 6.72 |
| | | Llama-3.2-3B | 1.00 | 8.00 | 0.87 | 9.07 | 0.68 | 6.27 | 0.83 | 8.17 | 0.34 | 3.44 | 0.74 | 6.99 |
| | | Llama-3.1-8B | 1.00 | 7.89 | 0.74 | 8.48 | 0.91 | 8.48 | 0.89 | 9.00 | 0.22 | 4.08 | 0.75 | 7.59 |
| | | Qwen3-1.7B | 1.00 | 8.11 | 0.86 | 9.02 | 0.60 | 6.06 | 0.72 | 6.22 | 0.20 | 3.62 | 0.68 | 6.61 |
| DisCIPL | Importance | Llama-3.2-1B | 0.89 | 5.56 | 0.79 | 7.68 | 0.62 | 5.06 | 0.78 | 7.89 | 0.20 | 3.26 | 0.66 | 5.89 |
| | | Llama-3.2-3B | 0.89 | 7.22 | 0.90 | 8.66 | 0.70 | 5.59 | 0.83 | 7.44 | 0.31 | 3.63 | 0.73 | 6.51 |
| | | Llama-3.1-8B | 0.89 | 7.33 | 0.78 | 8.02 | 0.91 | 7.33 | 0.78 | 7.39 | 0.21 | 3.76 | 0.71 | 6.77 |
| | | Qwen3-1.7B | 0.89 | 6.00 | 0.82 | 8.58 | 0.62 | 5.42 | 0.72 | 5.78 | 0.20 | 3.54 | 0.65 | 5.85 |
| DisCIPL | Rejection | Llama-3.2-1B | 0.89 | 7.67 | 0.91 | 9.07 | 0.48 | 8.84 | 0.78 | 9.17 | 0.01 | 7.64 | 0.62 | 8.48 |
| | | Llama-3.2-3B | 1.00 | 8.11 | 0.96 | 9.52 | 1.00 | 8.14 | 0.72 | 9.00 | 0.65 | 4.97 | 0.87 | 7.95 |
| | | Llama-3.1-8B | 1.00 | 8.78 | 0.96 | 9.46 | 0.97 | 8.69 | 0.89 | 9.17 | 0.73 | 4.80 | 0.91 | 8.18 |
| | | Qwen3-1.7B | 0.89 | 8.78 | 0.87 | 9.39 | 0.53 | 9.43 | 0.44 | 7.72 | 0.03 | 6.79 | 0.55 | 8.42 |
| DisCIPL* expert programs | SMC | Llama-3.2-1B | 0.89 | 8.11 | 0.86 | 9.37 | 0.98 | 8.92 | 0.94 | 8.61 | 0.56 | 4.13 | 0.85 | 7.83 |
| | Importance | Llama-3.2-1B | 0.89 | 8.11 | 0.98 | 9.17 | 0.97 | 7.78 | 0.78 | 7.33 | 0.49 | 3.42 | 0.82 | 7.16 |
| | Rejection | Llama-3.2-1B | 0.89 | 8.22 | 0.95 | 9.16 | 0.48 | 8.97 | 0.67 | 9.28 | 0.04 | 7.46 | 0.61 | 8.62 |
| Follower-only | Standard | Llama-3.2-1B | 0.88 | 6.89 | 0.68 | 8.98 | 0.10 | 8.81 | 0.24 | 9.22 | 0.00 | 7.53 | 0.38 | 8.28 |
| | | Llama-3.2-3B | 0.93 | 8.67 | 0.74 | 9.30 | 0.85 | 7.59 | 0.45 | 8.17 | 0.22 | 5.53 | 0.64 | 7.85 |
| | | Llama-3.1-8B | 0.99 | 8.33 | 0.79 | 9.46 | 0.84 | 8.44 | 0.68 | 8.89 | 0.34 | 4.82 | 0.73 | 7.99 |
| | Beam Search | Llama-3.2-1B | 0.89 | 7.11 | 0.67 | 9.40 | 0.03 | 9.97 | 0.32 | 9.50 | 0.00 | 8.81 | 0.38 | 8.96 |
| | | Llama-3.2-3B | 1.00 | 8.67 | 0.40 | 9.79 | 0.77 | 8.60 | 0.08 | 8.83 | 0.19 | 6.71 | 0.49 | 8.52 |
| | | Llama-3.1-8B | 1.00 | 9.00 | 0.30 | 9.80 | 0.90 | 8.89 | 0.71 | 8.94 | 0.25 | 5.58 | 0.63 | 8.44 |
| | Standard | Qwen3-1.7B | 0.71 | 8.67 | 0.68 | 9.34 | 0.10 | 9.41 | 0.17 | 8.56 | 0.01 | 6.83 | 0.34 | 8.56 |
| | Beam Search | Qwen3-1.7B | 0.69 | 8.67 | 0.65 | 9.56 | 0.01 | 9.68 | 0.13 | 8.39 | 0.02 | 6.29 | 0.30 | 8.52 |
| Planner-only | Standard | GPT-4o-mini | 0.99 | 9.44 | 0.88 | 9.86 | 0.73 | 9.81 | 0.66 | 9.50 | 0.33 | 6.91 | 0.72 | 9.10 |
| | | GPT-4o | 0.96 | 9.33 | 0.91 | 9.84 | 0.87 | 9.74 | 0.74 | 9.22 | 0.51 | 7.66 | 0.80 | 9.16 |
| | | GPT-4o (CoT) | 0.92 | 9.33 | 0.86 | 9.33 | 0.71 | 9.22 | 0.85 | 8.61 | 0.47 | 7.48 | 0.76 | 8.79 |
| Reasoning | Standard | o1 | 1.00 | 8.89 | 1.00 | 8.95 | 0.94 | 9.17 | 1.00 | 9.00 | 0.83 | 6.63 | 0.95 | 8.53 |

Table 4: **COLLIE Paragraph-Level Results.**

## C Puzzles Dataset and Results

The PUZZLES dataset consists of four task types with prompts given in Table 5. We re-run each method 10 times on each task type to produce the results in Table 6.

| TASK | PROMPT | | DISCIPL-SMC |
|------|--------|---|-------------|
| Square Poem | Write a poem with 8 lines, where each line has exactly 8 words. | ✓ | The sun sets slowly in the evening sky
Golden hues of orange and pink slowly fade
The stars come out one by one tonight
A night of rest for the world outside
The moon is full and bright and white
A beacon in the dark and endless night
The world is quiet and still and deep
In the stillness, all is calm and bright |
| Grant Proposal | Write an abstract for a grant proposal on elephant ecology and conservation. Your response should start with "Abstract: <YOUR ABSTRACT>". The abstract must be between 75-100 words. You are not allowed to use any of the following terms: conservation, sustainability, environment, ecology, wildlife, africa, asia, society, community, biodiversity, endangered, threatened, species, habitat, poaching, science, research. | ✓ | Abstract: Studying the complex behavior, social dynamics, and territorial boundaries of Indian elephant populations using camera trap images and GIS analysis reveals that strong elephant herds are consistently associated with increased access to food resources, particularly grasslands and aquatic habitats. These findings suggest that elephant social hierarchies play a crucial role in allocating resources and protecting against intrusions by buffalo herds. Investigating the spatial overlap between elephant and buffalo habitats, results show that elephants prioritize areas with larger buffer zones around their food sources to minimize the risk of competing utilization. |
| Ingredients List | Please write an ingredients list for chocolate chip brownies with at most 7 ingredients costing less than $18.00 total. The list should be in dashed bullet point format starting with "Ingredients:". Each ingredient should be listed on a separate line with the price given in USD. | ✓ | Ingredients:
- 1 and 1/2 sticks of unsalted butter ($3.50)
- 1 cup granulated sugar ($1.50)
- 2 large eggs ($1.50)
- 1/2 cup unsweetened cocoa powder ($1.50)
- 1 teaspoon vanilla extract ($1.25)
- 1 cup semi-sweet chocolate chips ($4.00) |
| Trip Itinerary | I'm planning a 3-day trip to Singapore. Please write me a detailed day-by-day itinerary that includes at least four activities per day. The itinerary should start with "Day 1:" and end with "Day 3:", with a blank line between each day. Each activity should be listed on a separate line starting with a time range in 24-hour format in square brackets (for example, "[11:00-13:00] Visit the Gardens by the Bay"). Make sure to leave at least 9 hours of free time each day for rest. | ✓ | Day 1:
[06:00-07:30] Start the day at Marina Bay Sands SkyPark for panoramic views of the city.
[07:30-09:30] Take a walk along the Marina Bay Waterfront Promenade for a leisurely stroll and enjoy the scenic views of the city.
[09:30-12:00] Visit the Esplanade - Theatres on the Bay for a movie or a performance.
[12:00-13:30] Have lunch at a nearby restaurant, such as Jumbo Seafood, and try some local seafood.
[13:30-15:30] Visit the Singapore Botanic Gardens for a relaxing walk and take in the beautiful gardens.
[15:30-18:00] Explore the iconic Gardens by the Bay, learn about Singapore's flora and fauna, and marvel at the stunning architecture.

Day 2:
[07:00-09:00] Start the day at the Singapore Zoo, home to over 2,800 animals from around the world.
[09:00-11:00] Visit the nearby Universal Studios Singapore for an immersive theme park experience with rides and attractions.
[11:00-12:30] Head to the iconic Merlion Park for a short walk and visit the Merlion statue.
[12:30-14:30] Have lunch at a local eatery, such as Maxwell Food Centre, and try some hawker food.
[14:30-16:30] Take a short taxi ride to the Gardens by the Bay for a visit to the Cloud Forest and Flower Dome.
[16:30-18:00] Relax and unwind at the Gardens by the Bay with a scenic walk.

Day 3:
[07:00-09:00] Visit the historic Fort Canning Park, a former British military base turned public park.
[09:00-11:00] Explore the nearby National Gallery Singapore, featuring a diverse collection of Southeast Asian art.
[11:00-12:30] Have lunch at a nearby eatery, such as Tiong Bahru, and try some Singaporean cuisine.
[12:30-14:30] Visit the nearby Little India and explore the vibrant streets of Chinatown and Little India.
[14:30-16:30] Take a short taxi ride to the iconic Marina Bay Sands and visit the rooftop bar for stunning views of the city. |

Table 5: **Puzzles tasks and example generations.** Samples were randomly selected from among the valid outputs produced by DISCIPL-SMC.

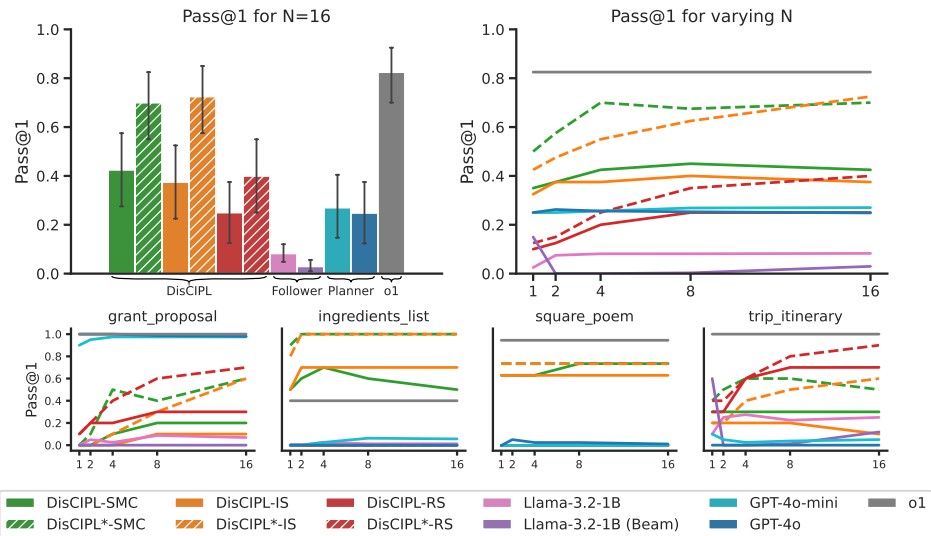

Figure 9: **Validity on PUZZLES.** Pass@1 for fixed *(top left)* and varying (top right) sample budgets for four challenge tasks *(bottom row)*. While DISCIPL still surpasses both Follower and Planner baselines, generation more often produces suboptimal programs, leading to a bigger gap vs. DISCIPL*.

| Method | Sampling | Model | Grant Proposal | | Ingredients List | | Square Poem | | Trip Itinerary | | Overall | |
| | | | Pass@1 | Coh. | Pass@1 | Coh. | Pass@1 | Coh. | Pass@1 | Coh. | Pass@1 | Coh. |
|---|---|---|---|---|---|---|---|---|---|---|---|---|
| DisCIPL | SMC | Llama-3.2-1B | 0.20 | 3.70 | 0.50 | 8.60 | 0.70 | 8.80 | 0.30 | 4.50 | 0.42 | 6.40 |
| | Importance | Llama-3.2-1B | 0.10 | 3.70 | 0.70 | 8.40 | 0.60 | 6.20 | 0.10 | 4.30 | 0.38 | 5.65 |
| | Rejection | Llama-3.2-1B | 0.30 | 9.00 | 0.00 | 8.90 | 0.00 | 9.10 | 0.70 | 8.40 | 0.25 | 8.85 |
| DisCIPL* | SMC | Llama-3.2-1B | 0.60 | 8.20 | 1.00 | 9.10 | 0.70 | 7.90 | 0.50 | 6.80 | 0.70 | 8.00 |
| | Importance | Llama-3.2-1B | 0.60 | 7.70 | 1.00 | 8.50 | 0.70 | 6.60 | 0.60 | 6.80 | 0.72 | 7.40 |
| | Rejection | Llama-3.2-1B | 0.70 | 8.80 | 0.00 | 8.60 | 0.00 | 9.30 | 0.90 | 7.90 | 0.40 | 8.65 |
| Follower-only | Standard | Llama-3.2-1B | 0.07 | 9.00 | 0.01 | 9.20 | 0.00 | 9.20 | 0.25 | 7.00 | 0.08 | 8.60 |
| | Beam Search | Llama-3.2-1B | 0.00 | 8.60 | 0.00 | 9.40 | 0.00 | 8.50 | 0.12 | 8.20 | 0.03 | 8.68 |
| Planner-only | Standard | GPT-4o-mini | 0.98 | 9.00 | 0.06 | 9.20 | 0.00 | 9.50 | 0.05 | 9.10 | 0.27 | 9.20 |
| | | GPT-4o | 0.98 | 9.00 | 0.00 | 9.80 | 0.01 | 9.50 | 0.00 | 9.40 | 0.25 | 9.43 |
| | | GPT-4o (CoT) | 0.91 | 9.00 | 0.17 | 9.50 | 0.01 | 9.40 | 0.12 | 9.20 | 0.30 | 9.28 |
| Reasoning | Standard | o1 | 1.00 | 9.00 | 0.40 | 8.80 | 0.90 | 8.90 | 1.00 | 9.30 | 0.82 | 9.00 |

Table 6: **PUZZLES Results.**

## D Weighted Pass@1

We formalize the the weighted Pass@1 metric used to evaluate model performance. This metric offers a natural way to incorporate sample-specific scores (e.g., particle weights $w_i$) into the evaluation. The approach is scale-invariant, so the absolute scale of the weights does not affect the metric.

Consider a set of $N$ samples, indexed by $i = 1, \ldots, N$. For each sample, we define:

- A log-probability $w_i \in \mathbb{R}$. In cases where $w_i$ is undefined (i.e., when generation produces a null output), we set its corresponding weight to zero.
- A binary pass indicator

$$I_i = \begin{cases} 1, & \text{if sample } i \text{ passes,} \\ 0, & \text{if sample } i \text{ fails.} \end{cases}$$

We define the unnormalized weight of sample $i$ as

$$\tilde{w}_i = \exp(w_i),$$

where for methods that do not compute sample weights, we assume uniform weighting by setting $\tilde{w}_i = 1$ for all $i$.

Weighted Pass@1 is defined as the probability that a single sample—drawn without replacement from the $N$ samples with probability proportional to $\tilde{w}_i$—is a passing sample. Formally, this is given by

$$\text{Weighted Pass@1} = \frac{\sum\limits_{i:I_i=1} \tilde{w}_i}{\sum\limits_{i=1}^{N} \tilde{w}_i}.$$

This expression represents the fraction of the total weight that corresponds to passing samples, and it reduces to the standard (unweighted) Pass@1 when all weights are equal.

## E Coherency Score

To assess coherency, we adopt the LLM-as-judge evaluation from Yao et al. (2024) (App. E). Whereas the original evaluation used GPT-4, to reduce inference time and cost, we use GPT-4o-mini (`gpt-4o-mini-2024-07-18`). We make minimal modifications to the original prompt to encourage GPT-4o-mini to adhere to the desired format.

```
Please concisely analyze the following text for coherency. Your analysis should be no
longer than 3 sentences and it *must* end verbatim with 'The coherency score is
<SCORE>', where <SCORE> is a number between 1 and 10.
```

## F Planner-Only Chain-of-Thought

### F.1 Chain-of-Thought Prompt

For the Planner-only (CoT) condition, we prompt GPT-4o with the following standard instructions to think step-by-step. These are provided as a system prompt.

```
You are helping a user generate text that satisfies constraints. Think step-by-step in
order to follow the user's instructions. Your final response should be enclosed in
<answer></answer> tags.
```

## F.2 Error Examples

### F.2.1 Character Counting

> **Task Prompt**  Please generate a sentence with exactly 82 characters. Include whitespace into your character count.

> **GPT-4o**  To create a sentence with exactly 82 characters, I'll count each letter, space, and punctuation mark:
>
> "The sun sets slowly over the ocean, painting the sky with hues of orange and pink delight."
>
> Now, let's count the characters:
>
> 1. "The sun sets slowly over the ocean, painting the sky with hues of orange" (70 characters)
> 2. " and pink delight." (12 characters)
>
> Total: 70 + 12 = 82 characters.
>
> <answer>The sun sets slowly over the ocean, painting the sky with hues of orange and pink delight.</answer>

> **Evaluator**  Invalid
> ✗  Length is 90 characters (expected 82)

### F.2.2 Word Positioning

> **Task Prompt**  Please generate a sentence:
> 1) with exactly 15 words;
> 2) with the 4th, 8th, 11th words to be 'collection', 'Noise', 'Testament' respectively.

> **GPT-4o**  To create a sentence that meets all the requirements, let's start by laying out the positions of the specific words:
>
> 1. (word 1)
> 2. (word 2)
> 3. (word 3)
> 4. collection
> 5. (word 5)
> 6. (word 6)
> 7. (word 7)
> 8. Noise
> 9. (word 9)
> 10. (word 10)
> 11. Testament
> 12. (word 12)
> 13. (word 13)
> 14. (word 14)
> 15. (word 15)
>
> Now, fill in the blanks with appropriate words to construct a coherent sentence:
>
> "The museum's vast collection included a fascinating exhibit titled Noise, featuring the Testament of artifacts."

> **Evaluator**  Invalid
> ✗  word 8 is "exhibit" (expected "Noise")
> ✗  word 11 is "featuring" (expected "Testament")

# G   DISCIPL Error Analysis

By learning from tracebacks, DISCIPL is able to correct roughly half of all initial runtime errors (Fig. 10, left). Nevertheless, even after 3 attempts, we observe error rates of 2-6% on COLLIE (App. B) and 18% on PUZZLES (App. C). Our analysis (Fig. 10, right) finds that most errors arise from invalid token masks and timeouts. Token mask errors occur when a call to observe() rules out all possible next tokens (e.g., when attempting to generate punctuation when a mask that rules out punctuation is already active). Meanwhile, timeout errors are more commonly associated with infinite loops or the omission of termination conditions in the step() logic.

There are also cases where bugs in the generated inference programs yield incorrect outputs without triggering any errors. In App. G, we give examples of an off-by-one error (Code Block 1) and an accidental infinite loop (Code Block 2) and highlight how structural issues with the outputs can be traced back to these bugs. In future work, we aim to implement a richer form of feedback where outputs from inference are provided to the Planner LM to correct for these kinds of errors.

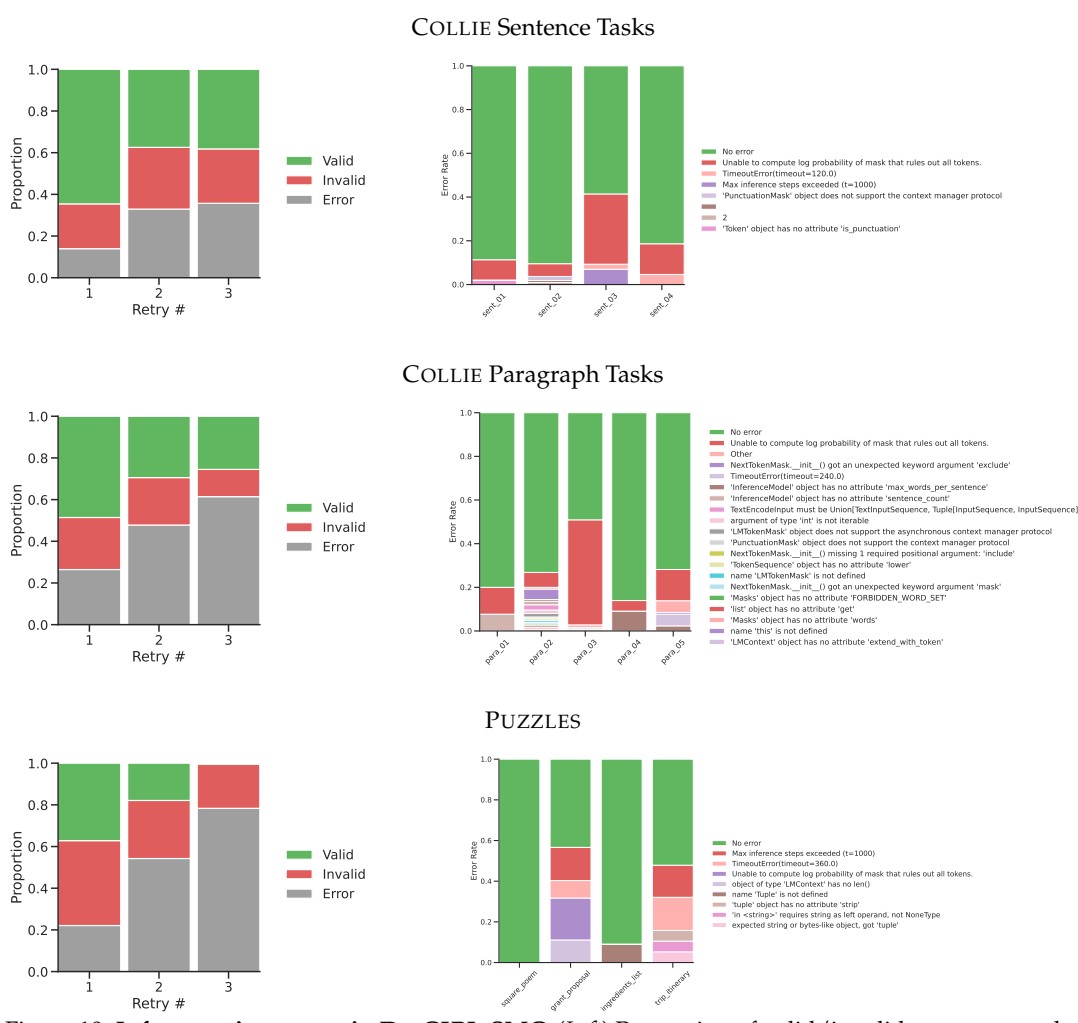

Figure 10: **Inference-time errors in DISCIPL-SMC.** (*Left*) Proportion of valid/invalid text generated at each retry, with errors in gray. Roughly half of all initial errors (#1) are corrected with feedback (#2), with diminishing returns on remaining errors for subsequent retries. (*Right*) Breakdown of Top-20 most frequent error messages aggregated across all retries.

**Listing 1** Example of a buggy program generated by the Planner LM for a COLLIE `sent_01` task instance. The generated program correctly applies `MaxCharsMask()` to produce a string that is exactly 102 characters. However, lines 33-34 subsequently force the Follower LM to generate punctuation, resulting in generations that exceed the target length by a single character.

```python
class InferenceModel(BaseModel):
    """Generates a sentence with exactly 102 characters, including whitespace."""

    def __init__(
        self,
        context,
        max_tokens: int = 32,
    ):
        super().__init__(
            context=context,
            max_tokens=max_tokens,
        )

        # Task-specific variables
        self.target_length = 102

    async def step(self):
        """
        Step granularity: token

        Generation strategy: At each step, sample a token that fits within the remaining length.
        Once the target length is reached, force punctuation and end the sentence.

        End condition: Target length is reached or token limit is reached.
        """

        # Sample a token that fits within the remaining length.
        async with MaxCharsMask(self, max_chars=self.target_length - len(self)):
            token = await self.next_token()

        # If we've reached the target length, force punctuation and end the sentence
        if len(self) == self.target_length:
            async with PunctuationMask(self):
                await self.next_token()
            await self.end()
            return

        # Enforce token limit
        if self.context.token_count > self.max_tokens:
            self.condition(False)
            await self.end()
            return
```

**Listing 2** Example of a buggy program generated by the Planner LM for a COLLIE para_05 task instance. The step() function contains a while loop that samples words until the target sentence-ending word is generated. However, due to a bug in the conditional at line 34, target words containing capital letters (e.g., Oxford, College) will never be matched. As a result, the Follower LM will generate the first two sentences of the paragraph correctly and then loop indefinitely on the third sentence until it exceeds max_tokens.

```python
class InferenceModel(BaseModel):
    """Generates a paragraph with exactly 5 sentences, where each sentence ends with a specific word."""

    def __init__(
        self,
        context,
        max_tokens: int = 256,
    ):
        super().__init__(
            context=context,
            max_tokens=max_tokens,
        )

        # Task-specific variables
        self.sentence_ends = ["offices", "door", "Oxford", "College", "initially"]
        self.sentence_count = 0

    async def step(self):
        """
        Step granularity: sentence

        Generation strategy: At each step, sample a sentence until the specific last word is reached.
        After generating the specific last word for a sentence, enforce punctuation.

        End condition: All sentences are generated or token limit is reached.
        """

        # Generate a sentence until the specific last word is generated
        target_last_word = self.sentence_ends[self.sentence_count]
        while True:
            word = await self.next_word()

            # Check if the generated word is the target last word
            if word.strip().lower() == target_last_word:
                # Force punctuation after the target last word
                async with PunctuationMask(self):
                    await self.next_token()
                break

            # Enforce token limit
            if self.context.token_count > self.max_tokens:
                self.condition(False)
                await self.end()
                return

        self.sentence_count += 1

        # End generation after all sentences are produced
        if self.sentence_count >= len(self.sentence_ends):
            await self.end()
            return
```

## H   Inference Methods

We provide a formal definition of the SMC algorithm in DISCIPL. Our implementation is adapted from the LLAMPPL with minor adjustments intended to provide robustness to bugs in autogenerated inference programs. In particular, we set a maximum number of SMC steps $T = 1000$ so that programs that fail to terminate do not run infinitely. We also invoke this method with a wall-clock timeout (not shown) to interrupt infinite loops that might occur internally within a single step().

---

**Algorithm 2** Sequential Monte Carlo Inference Algorithm

---

1: **function** SMC($\pi, M_F, N, \hat{N}_{\text{ess}}, T$)
2:   **for** $i = 1, \ldots, N$ **do**
3:    $x^{(i)} \leftarrow \pi(M_F)$           ▷ *Instantiate particles*
4:   **for** $t = 1, \ldots, T$ **do**
5:    *await* $x$.step() for $x$ in $\{x^{(1)}, x^{(2)}, \ldots, x^{(N)}\}$    ▷ *Advance particle state*
6:    **for** $i = 1, \ldots, N$ **do**
7:     $w^{(i)} \leftarrow \frac{w^{(i)}}{\sum_{j=1}^{N} w^{(j)}}$        ▷ *Normalize weights*
8:    **if** $\frac{1}{\sum_{i=1}^{N}(w^{(i)})^2} < \hat{N}_{\text{ess}}$ **then**    ▷ *Compute effective sample size*
9:     $a^{(i)} \sim \text{Categorical}\left(\frac{w^{(1)}}{\sum_{j=1}^{N} w^{(j)}}, \ldots, \frac{w^{(N)}}{\sum_{j=1}^{N} w^{(j)}}\right)$    ▷ *Resample*
10:     **for** $i = 1, \ldots, N$ **do**
11:      $(x^{(i)}, w^{(i)}) \leftarrow (x^{(a^{(i)})}, \frac{1}{N} \sum_{j=1}^{N} w^{(j)})$
12:    **if** $\forall x \in \{x^{(1)}, x^{(2)}, \ldots, x^{(N)}\}(x \in \mathcal{V}_{\text{EOS}}^*)$ **then**   ▷ *Check if all particles have terminated*
13:     **return** $\{x^{(1)}, x^{(2)}, \ldots, x^{(N)}\}$

---

## I   Implementation Details

LLAMPPL inference is performed with the vLLM backend, which uses PagedAttention (Kwon et al., 2023) for improved efficiency. For baselines, max_tokens is set to 32 for COLLIE sentences, 128 for COLLIE paragraphs, and 512 for PUZZLES. For DISCIPL, max_tokens is defined by the Planner LM and programs are executed with a max of $R = 3$ retries and a variable timeout (120s for sentences, 240s for paragraphs, and 360s for puzzles). The max sampling budget $N$ for each domain is determined based on the number of tasks and the generation length; we use $N = 32$ for sentences, $N = 8$ for paragraphs, and $N = 16$ for puzzles.

## J  Compute Cost Analysis

To help quantify the algorithmic tradeoffs of DISCIPL, we present an in-depth study of inference costs for a representative domain from our evaluations (COLLIE sentence tasks). Our analysis suggests that this is a conservative choice; comparative efficiency advantages of DISCIPL are likely even greater on longer-form generation tasks such as COLLIE paragraph generation and our PUZZLES domain.

We perform an analysis of the dollar cost of the DISCIPL-SMC method, the Follower-only, Planner-only, and o1 baselines.

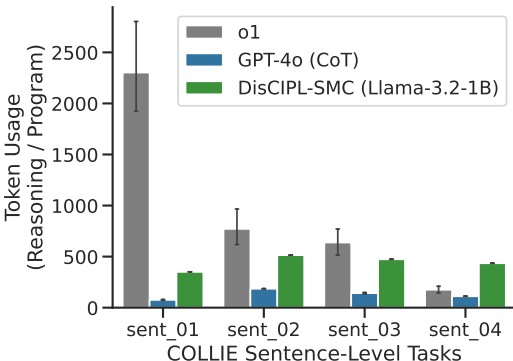

Figure 11: Comparison of the number of intermediate tokens generated by each method before producing an answer. DISCIPL programs are generally more compact than the reasoning traces generated by large reasoning models (e.g., o1) and more accurate than standard LLMs (e.g., GPT-4o with chain-of-thought).

**Key Takeaways**

- **In absolute dollar cost, DISCIPL is cheaper than o1:** As seen in Table 7, running DISCIPL is marginally cheaper than o1 ($4.26 vs. $4.75 per 100 tasks).

- **DISCIPL's inference programs are 40.1% shorter and 80.2% cheaper to generate than o1 reasoning traces.** Programs generated by the Planner LM use 40.1% fewer tokens than reasoning traces generated by o1. In addition, the generated tokens are on average 80.2% cheaper, as they are generated by a less expensive model—GPT-4o. Overall, per 100 tasks, the tokens of generated inference programs cost $0.91, whereas the tokens of o1 reasoning traces cost $4.59.

- **The cost of scaling inference-time compute with DISCIPL is negligible:** With 32 particles, the Follower LM's generations account for $< 0.1\%$ of the overall cost of DISCIPL. Thus, for example, doubling the test-time compute budget (e.g., from 32 to 64 particles) would have a negligible influence on the total cost of DISCIPL. By contrast, doubling the reasoning budget for o1 would nearly double its price, and similarly for Planner-only Chain-of-Thought.

- **In the current implementation, DISCIPL's cost is dominated by the long Planner system prompt:** Because GPT-4o has not been trained on LLaMPPL inference programs, we use in-context learning with a detailed system prompt and few-shot examples. We find that 72% of the cost of DISCIPL results from pre-filling the part of this prompt that is static across tasks.

- **Nevertheless, DISCIPL benefits significantly from prompt caching.** On average, 95.6% of the Planner's prompt tokens were cached by OpenAI's servers, resulting in an automatic 50% discount. In a performance-optimized system, these costs could be further reduced by using a local Planner LM with prefix caching. Alternatively, we could finetune the Planner LM on inference programs—in the same way that current reasoning models are finetuned on reasoning traces—to avoid the need for expensive re-prompting. Amortizing the Planner computation in this way would yield significant practical cost savings, up to 72% of the overall cost.

| | Answer | Reasoning / Program | Task Prompt | Planner Prompt (Non-Cached) | Planner Prompt (Cached) | Total |
|---|---|---|---|---|---|---|
| Follower-only (Llama-3.2-1B) | $0.0001 | - | $0.0003 | - | - | $0.0004 |
| Planner-only (GPT-4o) | $0.0345 | - | $0.0365 | - | - | $0.0709 |
| Planner-only CoT (GPT-4o) | $0.0376 | $0.2751 | $0.0485 | - | - | $0.3612 |
| o1 | $0.1061 | $4.5887 | $0.0509 | - | - | $4.7457 |
| DISCIPL-SMC* | $0.0032 | $0.9138 | $0.0003 | $0.2863 | $3.0804 | $4.2606 |

Table 7: Dollar cost per 100 tasks for each method.

**Column Descriptions:**

- **Answer:** The cost of generating the tokens of the solution to the task. For example, in o1, answer generation tokens are those generated after reasoning has completed; for GPT-4o CoT, they are the tokens tagged by the model as constituting the final answer; and in DISCIPL, they are all tokens generated as candidate answers by the Follower LM.

- **Reasoning / Program:** The cost of generating tokens instrumental to solving the task. For CoT and o1, these are reasoning tokens. For DISCIPL, these are the tokens of the inference program generated by the Planner LM.

- **Task Prompt:** The cost of pre-filling the task prompt, which describes the particular task being solved. For DISCIPL, this is the cost of explaining the task to the Follower, not to the Planner.

- **Planner Prompt:** The cost of pre-filling the Planner's system prompt in DISCIPL. Some of the prompt is task-specific and *not cached*, whereas some of the prompt is domain-general and *cached* across tasks, leading to a 50% discount under OpenAI's prompt caching policy.

## Token Counts and API Pricing Details

| | Input | Cached Input | Output |
|---|---|---|---|
| Llama-3.2-1B* | $0.06 | - | $0.06 |
| GPT-4o** | $5.00 | $2.50 | $20.00 |
| o1** | $15.00 | $7.50 | $60.00 |

Inference Cost for 1M Tokens (as of 2025-05-31)

*Follower LM compute actually performed with local GPU inference; costs shown use Together API pricing as a conservative upper bound. https://www.together.ai/pricing

**OpenAI model pricing; see https://platform.openai.com/docs/pricing, https://openai.com/index/api-prompt-caching/

**Average Token Counts and Standard Deviations per Task**

| | Answer | Reasoning / Program | Task Prompt | Planner Prompt (Non-Cached) | Planner Prompt (Cached) |
|---|---|---|---|---|---|
| Llama-3.2-1B | 14.9 (5.3) | - | 48.3 (2.0) | - | - |
| GPT-4o | 17.2 (4.5) | - | 72.9 (12.3) | - | - |
| GPT-4o (CoT) | 18.8 (25.7) | 137.6 (83.6) | 96.9 (12.3) | - | - |
| o1 | 17.7 (4.2) | 764.8 (1018.3) | 33.9 (12.3) | - | - |
| DISCIPL-SMC* | 23.2 (11.4) per particle | 456.9 (69.4) | 48.3 (2.0) | 12894.3 (444.1) | 12321.6 (1801.8) |

*DISCIPL-SMC instantiated with a Llama-3.2-1B Follower, GPT-4o Planner, $N = 32$ particles.

## K   Prompts and Code Links

### K.1   Planner System Prompt

We use a single system prompt written in the style of a `README.md` to demonstrate to the Planner LM how to write inference models. Since this prompt is written as a condensed explanation of language model probabilistic programming, it also doubles as a useful tutorial for the interested reader.

⊙  github.com/gabegrand/self-steering/blob/v1.0.0/disciple/prompts/planner_system_prompt.md

### K.2   Example Models

We define expert-written programs for the COLLIE task, which are appended to the Planner system prompt. Our experiments are leave-one-out design, such that the example(s) corresponding to the target task are always omitted from the prompt.

**COLLIE Example Models**

⊙  github.com/gabegrand/self-steering/blob/v1.0.0/evaluations/collie_eval/models.py

**PUZZLES Example Models**

⊙  github.com/gabegrand/self-steering/blob/v1.0.0/evaluations/puzzle/models.py

