# OpenReview forum: "Self-Steering Language Models"
_colmweb.org/COLM/2025/Conference — COLM 2025_

### Official Review · Reviewer_DG1U · 2025-05-09

**Rating:** 7
**Confidence:** 4
**Ethics Flag:** 1

**Summary:**

This paper proposes DisCIPL, a meta-reasoning framework that leverages a Planner LM to generate a probabilistic inference program that guides one or more Follower LMs to complete complex tasks. To empirically demonstrate the effectiveness of DisCIPL, the authors apply the proposed method to two constrained generation tasks (COLLIE and PUZZLES), with GPT-4o as the Planner and Llama-3.2-1B as the Follower. The method shows reasonable improvement over baselines (Planner-only and Follower-only) with respect to the weighted Pass@1 metric used to measure (constraint satisfaction) validity.

**Questions To Authors:**

1. While I understand the complicated probabilistic program is generated by the Planner via prompting, some statements and comments on the reproducibility and open-source plan of DisCIPL would be appreciated.

**Reasons To Accept:**

1. The proposed method (DisCIPL) is interesting and novel. I think it is a good instantiation to show how we might effectively combine the recurrent computation and parallel computation of LMs at test time.
2. The experimental design choices are reasonable to show the effectiveness of DisCIPL.The empirical results on COLLIE and PUZZLES make the method promising: the method enables a small LM (Llama-3.2-1B) to outperform GPT-4o and get close to reasoning LMs like o1.

**Reasons To Reject:**

1. I am concerned about the latency (in seconds) and scalability of this method, since it is sampling token-by-token (with sequential monte carlo). How much slower, if any, is the method compared to o1? How would the method scale to larger LMs, e.g. >= 7B?
2. It might be interesting to test the method on some tasks where coherence does not matter much, such as Game of 24, to make the empirical results more convincing and comprehensive (i.e. potential generalization to math reasoning).

---

> ### Author Response · Authors · 2025-06-03
> **Author response to Reviewer DG1U**
>
> Thank you very much for your positive review and recommendation for acceptance! We’re happy to hear that you found the experimental design choices reasonable to show the effectiveness of DisCIPL and that the empirical results seem promising.
>
> We appreciate the helpful feedback — in response to your suggestions, we’ve added experiments scaling up **[New Follower Models](https://openreview.net/forum?id=XvCBtm5PgF&noteId=VVy5FF0Kgh)** as well as a detailed **[Inference Cost Comparison](https://openreview.net/forum?id=XvCBtm5PgF&noteId=nkcaRtBzgM).**
>
> Please see responses to the points you raised below:
>
> ***How would the method scale to larger LMs, e.g. >= 7B?***
>
> We’ve added a range of new Follower models to our evaluations. These include scaling up Followers from the Llama-3.x family (to 3B and 8B, respectively), which we find improves DisCIPL’s performance. Please see the section on **[New Follower Models](https://openreview.net/forum?id=XvCBtm5PgF&noteId=VVy5FF0Kgh)** for more details.
>
> ***How much slower, if any, is the method compared to o1?***
>
> DisCIPL inference takes from a few seconds to several minutes, depending on the exact inference program, which is similar to o1’s inference speeds on our tasks. Nevertheless, in terms of other key metrics of inference cost, we find that DisCIPL programs are 40.1% shorter than o1 reasoning traces (in tokens) and are generated by a smaller model, meaning that DisCIPL is also less costly. (The dollar cost of running the inference program with a small Follower turns out to be negligible.) Please see our new **[Inference Cost Comparison](https://openreview.net/forum?id=XvCBtm5PgF&noteId=nkcaRtBzgM)** for more details.
>
> ***SMC requires sampling token-by-token; does this introduce overhead?***
>
> Thanks for this good question; there is some overhead introduced by SMC, but the reason this arises requires a bit of unpacking. First, note that standard autoregressive generation *also* involves sampling token-by-token from the LLM. However, in a standard (synchronous) decoding setting, each new token is a continuation of the same prompt prefix. In contrast, in our (asynchronous) decoding setting, we have multiple parallel processes issuing queries to the LLM, *each of which might have a different prompt prefix*. In a naive implementation, loading the different contexts and associated activations in and out of GPU memory introduces significant overhead. Fortunately, some really great tools have been developed recently to improve efficiency of serving LLMs in exactly this async setting. In our implementation, we use [vLLM](https://docs.vllm.ai/) as a backend, which implements features like KV caching and continuous batching that significantly improve inference efficiency. In practice, we find that vLLM results in a 2-5x speedup over the standard HuggingFace Transformers backend, making it possible to run SMC with many particles efficiently on a single A100/H100 GPU.
>
> ***It might be interesting explore generalization to math reasoning tasks (e.g., Game of 24)***
>
> We definitely agree and this direction (including Game of 24) is something that we had previously discussed among the authors. In brief, we agree that extending to math reasoning could help to make the case for the generality of the method. However, current approaches to math reasoning make certain assumptions (e.g., use of pre-trained reward models for tasks like MATH500 and AIME) that make it so that there is a single, “one-size-fits-all” inference strategy (i.e., calling the reward model periodically during generation, or just calling it once at the end to score completions). This is in contrast to the domains we look at, where there is meaningful variation in the inference programs on a task-by-task basis and the Planner really has to generalize (as opposed to just calling an off-the-shelf verifier). For a more in-depth discussion of some of these considerations, please see our response in **[Author responses to key issues](https://openreview.net/forum?id=XvCBtm5PgF&noteId=sGNLzrY7KV)**.
>
> Thanks again for the time spent in reviewing our submission and for your recommendation of acceptance — it means a lot to us! Please feel free to let us know if there’s anything further that you think we might be able to clarify for you or the other reviewers in the remainder of the rebuttal period.

---

> > ### Comment · Reviewer_DG1U · 2025-06-06
> >
> > Thanks to the authors for the detailed responses! I am happy to maintain my positive score.

---

### Official Review · Reviewer_WVvh · 2025-05-13

**Rating:** 6
**Confidence:** 3
**Ethics Flag:** 1

**Summary:**

This paper proposes to use generative models in a hierarchical framework of planned generations. A larger LLM produces a code making use of predefined tools, including smaller LLM calls, to generate an answer to a given query. This structure allows the framework to perform well at some tasks where direct LLM usage, even with CoT, would struggle (counting, constrained generation). It can also reduce inference cost and/or latency by using smaller models and parallel calls.

**Questions To Authors:**

Here are some suggestion about some building blocs to study more thoroughly:
The paper compares the DisCIPL approach to a reasoning model as O1, it seems that both could be combined and a reasoning model might produce better plans. Additionally, only a weak follower model (llama3.2-1b) has been tested. Many other models could be considered at no extra cost: Qwen3 0.6b, or 1.7b are both stronger and could be considered, they offer versions with thinking abilities that could also be interesting to consider; gemini-flash models are largely available for free and could be applied through API calls; gemma3 1b would also be a good option with much stronger abilities than llama3.2-1b (I am aware that some of these might not have been available during the writing of the paper but other models, stronger than llama3.2-1b, were available. I listed the current strong options at a similar cost).

The inference cost comparison is not clear. Table 1 shows better results for the o1 model, but they come at an extra cost, how can we compare the cost of DisCIPL that involves potentially several calls to the planner models (until no error) and many cheaper calls to the follower models? Maybe using only API models could work, for example using GPT4.1 for planning and GPT4.1-nano as follower and comparing with o3 on accuracy and total cost. Which one has the shortest latency to the final answer?

**Reasons To Accept:**

The proposed framework seems relevant for real-world applications, helping to reduce inference cost and latency while preserving or even improving the result quality.
Targeted experiments show some good features of the proposed approach.

**Reasons To Reject:**

Many building blocs are involved in this work and some of them are not thoroughly studied.

It seems that constraint satisfactions such as the one given in figure 1 are artificial tasks that are interesting as benchmarks to show features of LLMs capabilities but are not good performance indicators to be optimized. The proposed approach seems specifically good at solving this particular type of task but it is not convincing to show that the DisCIPL framework is more generally relevant. Puzzles tasks look somewhat more realistic but the dataset is very small. Showing more convincing use-cases seems important.

---

> ### Author Response · Authors · 2025-06-03
> **Author response to Reviewer WVvh**
>
> Thanks for the thoughtful review! We’ve added a number of additional experiments to the paper based on your suggestions, including experiments with **[New Follower Models](https://openreview.net/forum?id=XvCBtm5PgF&noteId=VVy5FF0Kgh)** as well as a new **[Inference Cost Analysis](https://openreview.net/forum?id=XvCBtm5PgF&noteId=nkcaRtBzgM)**. Please see responses below.
>
> ***Only a weak follower model (llama3.2-1b) has been tested. Many other models could be considered at no extra cost.***
>
> Thanks for your helpful and specific suggestions. We’ve added a range of new Follower models to our evaluations. These include scaling up Followers from the Llama-3.x family (to 3B and 8B, respectively) as well as new architectures, including your suggestion to use the Qwen3 architecture. Please see the section on **[New Follower Models](https://openreview.net/forum?id=XvCBtm5PgF&noteId=VVy5FF0Kgh)** for more details.
>
> As you predicted, we found some evidence that Qwen-3-1.7B is stronger than Llama-3.2-1B, both when evaluated by itself and also as part of DisCIPL.
>
> Regarding the other architectures you suggested:
>
> * **Gemma 3.** We also looked at extending these evaluations to include **Gemma-3-1B** as Follower. We ran into a small snag with vLLM compatibility (specifically pertaining to Gemma-3’s sliding window attention) that will require more time to address than we currently have due to COLM rebuttals. Nevertheless, we are excited about the possibility of including this architecture in our paper. Thanks again for the pointer to this model.
> * **Gemini Flash / GPT-4.1 / GPT 4.1-nano.** Thanks for these suggestions. One challenge we face with API models is that our method requires access to the Follower model’s logprobs in order to compute importance weight updates (this is described in S3.1 of the paper under “General Framework”). Additionally, to speed up inference, we use vLLM as a backend for KV-caching, which also requires the Follower to be run locally. That said, note that in principle DisCIPL-RS could be run without logprobs, and thus could be instantiated with GPT 4.1-nano or other API models as Followers. In our initial experiments, we found DisCIPL-RS to be weaker than DisCIPL-SMC, but some findings from our **[New Follower Models](https://openreview.net/forum?id=XvCBtm5PgF&noteId=VVy5FF0Kgh)** experiments suggest that for stronger Follower models, DisCIPL-RS can be competitive.
>
> ***The inference cost comparison is not clear.***
>
> We appreciate the feedback; this was also something that other reviewers noted. Please see our new **[Inference Cost Comparison](https://openreview.net/forum?id=XvCBtm5PgF&noteId=nkcaRtBzgM)** section. We have refined our analysis to provide a clear apples-to-apples dollar cost comparison using current API pricing — we hope this helps to address your questions.
>
> ***Constrained generation tasks are artificial; showing more convincing use-cases seems important.***
>
> Thanks for this feedback. We definitely believe our method could also be extended to other domains (e.g., math reasoning) and that this could strengthen the narrative. Given the limited time in the rebuttal period, we’ve chosen to focus for now on addressing empirical questions raised by reviewers that can be productively addressed within the scope of the existing domains. Please see our responses in **[Author responses to key issues](https://openreview.net/forum?id=XvCBtm5PgF&noteId=sGNLzrY7KV)** for more discussion on this question.
>
> Thank you again for your helpful suggestions regarding the Follower model experiments and inference costs comparisons! Please don’t hesitate to let us know if you have any follow-up questions that arise from looking at these new results. We’re grateful for your time and consideration.

---

> ### Author Response · Authors · 2025-06-06
>
> Dear Reviewer,
>
> As we near the end of the discussion period, we wanted to make sure you had a chance to take a look at the rebuttal we posted earlier this week.
>
> We ran several new experiments that specifically address your comments. We hope you will consider these in your final evaluation, and we’d be grateful for any further feedback or follow-up questions you might have.
>
> Thank you again for your time and consideration.
>
> Sincerely,
>
> The Authors

---

> ### Comment · Reviewer_WVvh · 2025-06-07
>
> Except for more convincing benchmarks, all my concerns and questions were answered. I will raise my score accordingly and would have raised even more given more benchmarks. Maybe something like SWE bench could be considered. I would like to point out that I am generally not convinced by arguments claiming that other papers use the same standard of evaluation quality. This goes against progress and only sets your work as part of theses lower standard of benchmarking instead of raising it to a convincing level of scientific evidence.
> That being said I appreciate the extensive set of new experiments that the authors brought to the rebuttal.

---

> > ### Author Response · Authors · 2025-06-07
> >
> > Thank you kindly for your response! We’re glad that our new experiments were able to address many of your concerns/questions -- **we really appreciate your vote for acceptance.**
> >
> > We agree that code generation (e.g., SWE-bench) could be an interesting evaluation for this approach. While we would have liked to explore other benchmarks, due to the compressed timeline of the rebuttal period, we weren’t able to produce full evaluation results on a new domain. We’d emphasize that, in comparison to other common research artifacts (e.g., a new Transformer model release) that are plug-and-play with many evaluation harnesses, the multi-LLM aspect of our method means that more effort is required to produce new evaluations.
> >
> > We will note that we’ve been running some preliminary experiments on the [Google IFEval dataset](https://arxiv.org/abs/2311.07911) [1], which is part of the [EleutherAI LM Evaluation Harness](https://github.com/EleutherAI/lm-evaluation-harness). While IFEval is also an instruction-following / constrained generation benchmark, the task instructions are more naturalistic than COLLIE. Our initial results show some promising evidence of zero-shot transfer: we find that even with no additional example programs for this dataset, DisCIPL with GPT-4o/Llama-3.2-1B can generate correct programs and outputs for many IFEval tasks. Although we don’t currently have full results ready to share on IFEval, this is something we could prep in time for the COLM camera-ready if the reviewers believe an additional benchmark is of interest.
> >
> > In the meantime, we’ve directed our efforts during rebuttal towards providing the reviewers with empirical data to address questions regarding scaling properties, inference costs, and baselines through the lens of our existing domains. We appreciate your acknowledgment of the time and effort that went into producing these new experiments. The comparison to ToT/SoS is intended to highlight that in its current state, the evaluation scope is commensurate with other recent works accepted to COLM, NeurIPS, etc.
> >
> > Thank you again for your engagement during the rebuttal period and for your updated score. We wish you all the best.
> >
> > ---
> > [1] Jeffrey Zhou, Tianjian Lu, Swaroop Mishra, Siddhartha Brahma, Sujoy Basu, Yi Luan, Denny Zhou, and Le Hou. "Instruction-following evaluation for large language models." *arXiv preprint arXiv:2311.07911* (2023).

---

### Official Review · Reviewer_3m39 · 2025-05-15

**Rating:** 5
**Confidence:** 4
**Ethics Flag:** 1

**Summary:**

This paper proposes DisCIPL, an agent framework including a planner and a follower. The planner writes a program about calling LMs to do generation (asynchronously), and the follower follows the program to generate. They show that a weak follower can achieve good performace.

**Questions To Authors:**

SMC seems to be not necessary here. Can we use async probabilistic programming with standard parallel repeating generation?

**Reasons To Accept:**

The framework incorporates agent workflow, asynchronous probabilistic programming, sequential Monte Carlo, which seems to be interesting.

**Reasons To Reject:**

1. The approach doesn't seem very practical. DisCIPL relies on a strong model to act as the planner and generate code, which is the most critical step. If such a strong model is already available, it's unclear why weaker models are used in subsequent stages.

2. The paper lacks an appropriate baseline. If we only consider using the strong model as illustrated above, the setup closely resembles PoT-style methods. In that case, comparisons should be made against other agent-based workflows, rather than simply evaluating the strong model alone.

3. The benchmark used appears to have already been solved by earlier reasoning models (e.g., o1). This raises the possibility that current techniques may easily solve the tasks presented in the paper.

4. The approach may not generalize well to other types of planning problems.

---

> ### Author Response · Authors · 2025-06-03
> **Author response to Reviewer 3m39**
>
> Thanks for your insightful review! In response to your suggestions, we’ve added several new experiments, including additional **[Planner Baselines: CoT and PoT](https://openreview.net/forum?id=XvCBtm5PgF&noteId=h9DYKaqMkY)** as well as a new **[Inference Cost Analysis](https://openreview.net/forum?id=XvCBtm5PgF&noteId=nkcaRtBzgM)**.
>
> ***The setup closely resembles Program-of-Thought (PoT) style methods; can you add a baseline?***
>
> Thank you for the suggestion! Your point is well taken that if we assume access to a strong Planner model (e.g., GPT-4o) then there are other ways of using that model besides the existing Planner-only baselines, which perform direct generation.
>
> In response to your feedback, we explored two different approaches that utilize the Planner in more computationally-intensive ways:
>
> * **Planner Chain-of-Thought (CoT)**
> * **Planner Program-of-Thought (PoT)**
>
> We provide a detailed writeup of these experiments **[in a separate post: Planner Baselines: CoT and PoT](https://openreview.net/forum?id=XvCBtm5PgF&noteId=h9DYKaqMkY)**.
>
> Regarding PoT specifically, one of our co-authors spent quite a bit of time doing prompt engineering to try to get PoT to work for our domains; [**you can read the writeup from this exploration here**](https://self-steering.notion.site/Planner-Program-of-Thought-PoT-Experiments-206e3927c7e180aa9274ec0a3eb22057).
>
> Overall, we found that it’s quite difficult to generate vanilla Python programs to solve the kinds of tasks in our evaluation: Whereas the original PoT paper was designed for numerical reasoning tasks, our work is focused on text generation. (This is why our programs are designed to utilize Follower LMs, which are good at generating text). For this reason, and due to time limitations, we ended up not running full experiments with PoT; nevertheless, if after reading you have suggestions for overcoming these hurdles, we would be happy to continue to pursue this direction.
>
> ***If such a strong model is already available, it's unclear why weaker models are used in subsequent stages.***
>
> Good question — we show that there are multiple reasons why we would want to offload computation to Follower LMs.
>
> * **Performance:** We find that the Planner alone actually doesn't perform that well on many tasks (see Planner-only baselines, including the new CoT baseline above). For instance, on COLLIE sentence tasks, GPT-4o only gets 24% Pass\@1, compared to 81-87% for DisCIPL. This highlights a key point of this paper: We can use a relatively strong (but still unreliable) LM as a Planner and far surpass its performance with our method.
> * **Scalability:** We can efficiently run inference with many Followers, whereas scaling up the Planner LM is orders of magnitude more expensive. Specifically, the marginal cost of doubling inference-time compute in DisCIPL w/ Llama-3.2-1B is *100x cheaper than GPT-4o* and *10000x cheaper than o1*. Even when we factor in the cost of generating inference programs with a GPT-4o Planner, we find that our approach is still less expensive than o1. (Please see our section on **[Inference Cost Analysis](https://openreview.net/forum?id=XvCBtm5PgF&noteId=nkcaRtBzgM)** for more details.)
>
> ***The benchmark is already solved by o1.***
>
> Our primary argument here is about efficiency, not absolute performance. Please see the section, “What is the role of o1 in the experiments?” in **[Author responses to key issues](https://openreview.net/forum?id=XvCBtm5PgF&noteId=sGNLzrY7KV)**.
>
> ***The approach may not generalize well to other types of planning problems.***
>
> While we’re not sure what problems you might have in mind, we believe our approach *could* generalize to many kinds of planning problems that have a natural step-by-step decomposition. In particular, we think the ideas from DisCIPL might apply to agentic tasks like web navigation; please see our **response to Reviewer DTJV** where we explored this idea.
>
> **SMC seems to be not necessary here. Can we use async probabilistic programming with standard parallel repeating generation?**
>
> Yes! **This condition is already included in our paper as DisCIPL-RS** (see Lines 176-179 for a description and Table 1 / Figs. 3, 5, 7, 8 for results). Note that this variant of our method uses the same probabilistic programs to perform standard parallel repeating generation (aka rejection sampling) as you described. We find that **this approach consistently underperforms SMC by a margin of 9%-70%** (absolute percentage points) across domains.
>
> We hope these responses answered your questions. Thanks for your time and effort in engaging with our submission. Please don’t hesitate to follow up if there is anything further we might clarify. Thank you for your consideration!

---

> ### Author Response · Authors · 2025-06-06
>
> Dear Reviewer,
>
> As we near the end of the discussion period, we wanted to make sure you had a chance to take a look at the rebuttal we posted earlier this week.
>
> We ran several new experiments that specifically address your comments. We hope you will consider these in your final evaluation, and we’d be grateful for any further feedback or follow-up questions you might have.
>
> Thank you again for your time and consideration.
>
> Sincerely,
>
> The Authors

---

> ### Author Response · Authors · 2025-06-10
> **Awaiting follow-up; discussion closes today**
>
> Dear Reviewer 3m39,
>
> As the discussion period concludes today, we wanted to follow up to see if you’ve had a chance to review the updates on the thread. We greatly appreciate your detailed review and suggestions, which have guided several new additions, including:
>
> - [Planner Baselines: CoT and PoT](https://openreview.net/forum?id=XvCBtm5PgF&noteId=h9DYKaqMkY)
>
> - [Inference Cost Analysis](https://openreview.net/forum?id=XvCBtm5PgF&noteId=nkcaRtBzgM)
>
> We're grateful for the constructive engagement from all reviewers—two have updated their scores, and all three have now recommended acceptance.
>
> If you have time today, we’d be grateful to hear whether these updates address your concerns. We're also happy to provide any further clarification or answer any remaining questions that you may have.
>
> Sincerely,
>
> The Self-Steering Authors

---

### Official Review · Reviewer_DTJV · 2025-05-23

**Rating:** 6
**Confidence:** 5
**Ethics Flag:** 1

**Summary:**

This paper introduces DisCIPL, a novel meta-reasoning framework designed to enhance inference in language models (LMs) through "self-steering." The core innovation involves decomposing the inference process into a Planner model, which generates a task-specific probabilistic inference program, and Follower models, which execute these inference procedures. By utilizing recursive search procedures and leveraging highly parallelizable Monte Carlo methods, DISCIPL aims to combine the flexibility of chain-of-thought reasoning with the efficiency and verifiability of structured inference techniques. A significant contribution is the demonstrated capacity of smaller LMs, when integrated into the DisCIPL framework, to achieve performance comparable to larger, state-of-the-art models on complex constrained generation tasks.

Empirically, DisCIPL shows improvements in performance over baseline models across two benchmarks: COLLIE, a constrained text generation dataset, and PUZZLES, a diverse set of naturalistic tasks. Notably, when instantiated with the relatively small Llama-3.2-1B as the Follower, DisCIPL's performance matches and sometimes surpasses that of larger models like GPT-4o and o1, particularly in tightly constrained generation tasks. The empirical analysis further highlights that Sequential Monte Carlo (SMC) inference provides better coherence and constraint satisfaction than other methods, validating the efficiency and effectiveness of the proposed inference programming approach.

**Questions To Authors:**

- A.10 system prompt is missing
- In addition to providing such a system prompt, can the authors run an ablation study (e.g., prompt variants, few‑shot vs. zero‑shot) to clarify how sensitive DisCIPL is to prompt design?
- We might want to explore stronger or weaker planners (e.g., GPT‑4‑Turbo vs. GPT‑3.5) and a broader range of follower sizes (e.g., 7 B, 13 B) to establish clearer scaling trends and performance ceiling
- A detailed breakdown of compute cost for DisCIPL vs. baseline large models (and basic sampling methods like beam search) would help practitioners judge real‑world trade‑offs.
- Can we adapt DisCIPL to broader agentic settings (e.g., multi‑step tool use, web navigation)? Outlining concrete next steps or preliminary experiments would strengthen the case for wider impact.

**Reasons To Accept:**

1. DisCIPL advances LLAMPPL by automating the generation of inference programs (with options of sampling methods and common patterns). Allowing a Planner LLM (e.g., GPT-4o) to synthesize LLAMPPL programs autonomously is an interesting methodological innovation.
2. Separating planning (high-capability LLM) from execution (smaller, computationally lighter LLM) allows smaller, more resource-efficient models to achieve performance comparable to larger models.
3. The introduction and utilization of PUZZLES show some efforts in generalizing the problem and proposed method to a relatively more realistic setting

**Reasons To Reject:**

1. The automated generation of inference programs heavily relies on a system prompt (missing in the current version, possibly due to negligence in formatting). Such a prompting method is not rigorously tested in the current version, and we have less clue on how and why it works, leaving unclear the underlying mechanisms responsible for the reported successes.
2. While the PUZZLES testing set attempts to introduce more realistic scenarios, the overall problem scope remains narrowly confined to constrained-generation tasks. The method, as presented, does not adequately explore broader, potentially more impactful applications. I can imagine we may extend this into an agentic framework for a more powerful application scope.
3. The surprisingly strong performance of baseline models (like o1) on challenging reasoning tasks suggests that general long-form reasoning might be able to generalize well to the problems this paper tried to solve. On the other hand, the proposed method essentially tries to inject more explicit priors to guide small LLMs, which seems not generalizable (and also powerful) as o1.
4. The absence of diverse baselines limits the robustness of empirical claims (we may need more serious baselines). Moreover, the evaluation currently relies solely on a single planner-follower pair. Additional experiments involving stronger planners and followers could clarify the method’s scalability and show its comparative advantage over existing state-of-the-art reasoning models (outperforming o1 seems be another necessary milestone )

---

> ### Author Response · Authors · 2025-06-03
> **Author response to Reviewer DTJV**
>
> Thank you for this very insightful review. In response to your feedback, we’ve added a number of new experiments, including **[New Follower Experiments](https://openreview.net/forum?id=XvCBtm5PgF&noteId=VVy5FF0Kgh)**, **[New Planner-only Baselines](https://openreview.net/forum?id=XvCBtm5PgF&noteId=h9DYKaqMkY)**, and a new **[Inference Cost Analysis](https://openreview.net/forum?id=XvCBtm5PgF&noteId=nkcaRtBzgM)**.
>
> ***Additional experiments involving stronger Planners and Followers could clarify the method’s scalability.***
>
> Thank you very much for the suggestion. We agree this is an interesting direction, particularly the question of how performance scales with larger Follower LMs. In response to your suggestion, we ran several **[New Follower Experiments](https://openreview.net/forum?id=XvCBtm5PgF&noteId=VVy5FF0Kgh)**.
>
> Briefly, we find that DisCIPL scales with Follower size (+0.09 Pass\@1 with 8B vs. 1B Llama) and benefits from stronger small LMs like Qwen3-1.7B (+0.07 Pass\@1 over 1B Llama). We also observe some other interesting scaling trends regarding the validity/coherency tradeoff.
>
> Regarding the Planner, we also ran some new experiments with stronger variants of the GPT-4o Planner-only baselines (**[see here](https://openreview.net/forum?id=XvCBtm5PgF&noteId=h9DYKaqMkY)**).
>
> We also agree with your comment that scaling *down* the Planner within the context of DisCIPL would be an interesting direction to explore (though it seems that not all the reviewers would be in agreement on this direction). In particular, while we didn’t have time in the current rebuttal push to run experiments with weaker Planners, we think it would be interesting if there were some intermediate-sized LLM (e.g., one of the DeepSeek-R1 distills) that were capable of acting as both Planner and Follower. We leave this for future work!
>
> ***A detailed breakdown of compute cost would help practitioners judge real‑world trade‑offs.***
>
> Thanks as well for this suggestion; we definitely agree. We've added a detailed **[Inference Cost Analysis](https://openreview.net/forum?id=XvCBtm5PgF&noteId=nkcaRtBzgM)**, which we think will be of interest. Please don’t hesitate if you have any further questions.
>
> ***Outperforming o1 seems to be another necessary milestone***
>
> Please see the section, “What is the role of o1 in the experiments?” in **[Author responses](https://openreview.net/forum?id=XvCBtm5PgF&noteId=sGNLzrY7KV)**.
>
> ***The proposed method essentially tries to inject more explicit priors to guide small LLMs, which seems not generalizable (and also powerful) as o1***
>
> We’re not sure we agree with this characterization; this paper is about a general framework that decomposes reasoning into a two-step process with Planner LMs and Follower LMs. We emphasize that these roles can be played by arbitrary models, and that we could, in principle, use o1 *either* as a Planner or Follower. While certain resource limitations prevent us from doing so at the moment (see **[Author responses](https://openreview.net/forum?id=XvCBtm5PgF&noteId=sGNLzrY7KV)** for a discussion), we would love to see a group with the relevant resources explore instantiations of DisCIPL that use large-scale reasoning models like o1 in both Planner and Follower roles.
>
> ***The system prompt is missing from A.10.***
>
> Thanks so much for spotting this issue — we agree this is an important piece of context for reviewers and we sincerely apologize for the inconvenience.
>
> (By way of explanation, the system prompt is written in Markdown and we were using a package to automatically compile it to LaTex; it appears that the final compilation of the submission PDF caused an error at the last minute that we didn’t catch.)
>
> While authors are prevented from updating the PDF during COLM rebuttal, **we’ve uploaded the full [System Prompt](https://self-steering.notion.site/Planner-System-Prompt-203e3927c7e1801e97cff7e4c12cd487) for your reference**.

---

> > ### Author Response · Authors · 2025-06-03
> >
> > ***Can the authors run an ablation study (e.g., prompt variants, few‑shot vs. zero‑shot) to clarify how sensitive DisCIPL is to prompt design?***
> >
> > We agree that ideally, performance shouldn’t be too sensitive to the exact prompt. Regarding prompt ablations, since the Planner LM is not assumed to be familiar with LLaMPPL out-of-the-box, we expect limited effectiveness from zero-shot prompting. (This is also why the system prompt is quite long.) That said, we might be able to use the existing system prompt to demonstrate zero-shot transfer to another domain—while we didn’t have time to add any new domains for the rebuttal, this is definitely something we’re interested in exploring.
> >
> > In the meantime, another way of looking at this design choice is that we’re using in-context learning as a substitute for learning a Planner LM (e.g., via finetuning) that knows how to write LLaMPPL code. The main advantage of ICL in a research setting is that this enables rapid prototyping. Time permitting, we’d be happy to run any specific ablations that come to mind on review of the [system prompt linked above](https://self-steering.notion.site/Planner-System-Prompt-203e3927c7e1801e97cff7e4c12cd487).
> >
> > ***Can we adapt DisCIPL to broader agentic settings (e.g., multi‑step tool use, web navigation)?***
> >
> > Great question — one application that comes to mind is that in many agentic settings, such as web navigation, we run into issues where an agent makes an incorrect decision (e.g., clicking on the wrong link) and get stuck going down a garden path **[e.g., 1, 2]**. These situations can be difficult to recover from (the agent needs to realize that it made a mistake and backtrack) and also significantly wasteful (the agent consumes substantial tokens exploring unpromising paths).
> >
> > In these settings, one way of overcoming these issues would be to decompose the agentic task following the pattern we introduce in DisCIPL.
> >
> > - **First, the Planner model** would produce a high-level plan for solving the task, along with some kind of scoring/verification criterion (*Did the agent visit the right URL in a webnav setting? Did the generated code pass the unit tests in a SWE setting?*). Importantly, these scoring criteria can be continuous/soft (*what percentage of the test cases were passed?*) and can also apply to intermediate states (*does the current website seem like it will lead to an answer to the user’s query, as determined by an LLM-as-judge?*).
> > - **Next, the Follower models** (acting as agents) would explore various action trajectories in parallel, periodically pausing at intermediate steps to report results to the Planner model. We can then compute importance weight updates based on the Follower scores and perform SMC resampling.
> >
> > The proposed setup would have the effect of dynamically reallocating compute from agents exploring unpromising trajectories, towards other agents who are on more promising paths, with the potential to reduce resources wasted on unnecessary exploration.
> >
> > This is just a sketch of how the ideas from our work in DisCIPL could be applied to an agentic setting. Realistically, there are many details that would need to be worked out that would likely represent a whole new paper’s worth of research. That said, we hope this helps to address your questions — thank you for the insightful question and the opportunity to brainstorm about possible extensions of our work!
> >
> > Finally, we’d like to thank you again for your time and effort in engaging with our submission. Please don’t hesitate to follow up if there is anything further we might clarify that would raise your score. Thanks for your consideration!
> >
> > **References**
> >
> > **[1]** Shikhar Murty, Hao Zhu, Dzmitry Bahdanau, and Christopher D. Manning. "NNetNav: Unsupervised learning of browser agents through environment interaction in the wild." In *Scaling Self-Improving Foundation Models without Human Supervision*. 2024.
> >
> > **[2]** Zhisong Zhang, Tianqing Fang, Kaixin Ma, Wenhao Yu, Hongming Zhang, Haitao Mi, and Dong Yu. "Enhancing Web Agents with Explicit Rollback Mechanisms." *arXiv preprint arXiv:2504.11788* (2025).

---

> ### Author Response · Authors · 2025-06-06
>
> Dear Reviewer,
>
> As we near the end of the discussion period, we wanted to make sure you had a chance to take a look at the rebuttal we posted earlier this week.
>
> We ran several new experiments that specifically address your comments. We hope you will consider these in your final evaluation, and we’d be grateful for any further feedback or follow-up questions you might have.
>
> Thank you again for your time and consideration.
>
> Sincerely,
>
> The Authors

---

> > ### Comment · Reviewer_DTJV · 2025-06-10
> > **Thanks for the rebuttal**
> >
> > I thank the authors for their thorough and thoughtful response, particularly for addressing several key concerns with new experiments and clarifications. The additional follower experiments, insights into scaling behavior, and the comprehensive breakdown of inference costs notably strengthen the empirical foundation of the paper. Moreover, the authors’ discussion about extending DisCIPL into broader agentic tasks offers promising future directions that could significantly enhance the method's applicability. Although some questions remain, especially around sensitivity to prompt design and broader scalability across various planners and tasks, the detailed responses provided have sufficiently addressed my primary concerns. Consequently, I've decided to increase my score to reflect this improvement.

---

> > > ### Author Response · Authors · 2025-06-10
> > > **Thank you for your support!**
> > >
> > > Thank you again for your thorough review and constructive engagement during the rebuttal period. We sincerely appreciate your recognition of our new experiments, including additional follower evaluations, scaling analyses, planner baselines, and the detailed inference cost breakdown. Your insightful feedback was instrumental in guiding these improvements and significantly enhancing the empirical strength of our paper.
> > >
> > > We also found the exchange regarding the broader applicability of DisCIPL to agentic tasks particularly stimulating, and it has inspired promising future directions for our work.
> > >
> > > Your decision to raise your score means a great deal to us, and we are grateful for your support and recommendation for acceptance.

---

### Author Response · Authors · 2025-06-03
**Author Response: Overview**

Thank you to the reviewers for your valuable time in considering our submission. We sincerely appreciate your insightful comments, suggestions, and constructive feedback.

### Highlights from Reviewer Comments

We’re grateful for the many positive comments from the reviewers highlighting strengths of our work.

- **Methodological innovation and intellectual appeal:** Reviewers broadly found self-steering to be a “novel” and “interesting” technique for leveraging parallel LM computation via probabilistic inference (DTJV, DG1U, 3m39, WVvh).
- **Strong efficiency and performance:** Reviewers cited “promising” empirical results that “show the effectiveness of DisCIPL,” noting that our method enables a 1B Llama model to outperform GPT-4o and approach the performance of o1 (DTJV, DG1U, WVvh).
- **Potential impact for production systems:** Because the framework can lower inference cost and is naturally parallel, reviewers believe our work could be “relevant for real-world applications” (WVvh).

### Summary of Key Reviewer Concerns

Below, we summarize the areas of concern noted by different reviewers.

**Broadening evaluation of Follower model sizes and families.** Reviewers DTJV and WVvh expressed a desire to see instantiations of DisCIPL using LMs beyond the existing Planner/Follower pair of GPT-4o/Llama-3.2-1B.

→ **We’ve added new experiments on [Scaling Follower model size and diversity](https://openreview.net/forum?id=XvCBtm5PgF&noteId=VVy5FF0Kgh).**

**Stronger Planner baselines.** Reviewer 3m39 requested stronger variants of the existing Planner-only baseline; these included chain-of-thought (CoT) and program-of-thought (PoT) techniques with the Planner LM.

→ **We report findings from experiments with new [Planner-only baselines](https://openreview.net/forum?id=XvCBtm5PgF&noteId=h9DYKaqMkY).**

**Compute cost comparisons.** Reviewers DTJV, WVvh, and DG1U asked for more detailed comparisons of the computational cost of DisCIPL vs. baselines to clarify the efficiency argument.

→ **We’ve produced a detailed [Inference cost comparison](https://openreview.net/forum?id=XvCBtm5PgF&noteId=nkcaRtBzgM).**

**Scope of evaluation domains.** Some reviewers felt the evaluations, which focused primarily on constrained generation tasks, were too restricted in scope. Reviewer DTJV noted that, while the PUZZLES domain points towards applications to more real-world tasks, they were curious about potential applications to more agentic settings like web navigation or tool use. Meanwhile, Reviewer DG1U asked about applications to math reasoning tasks like the Game of 24 / Countdown.

→ **Please see the discussion in [Author responses to key issues](https://openreview.net/forum?id=XvCBtm5PgF&noteId=sGNLzrY7KV)**.

**Other concerns.** We address additional issues raised by reviewers, including reproducibility, the Planner system prompt, and o1 performance comparisons, in [Author responses to key issues](https://openreview.net/forum?id=XvCBtm5PgF&noteId=sGNLzrY7KV).

---

### Author Response · Authors · 2025-06-03
**Overview of New Experiments and Analysis**

As much as possible, we’ve aimed to address reviewer comments empirically through new experiments and analysis. Our experiments during rebuttal have produced a significant amount of new data, including 9 new DisCIPL conditions and 7 new baselines.

For skimmability, we’ve put together three new sets of results:

*Key findings below highlight results from COLLIE sentence-level tasks; please see the individual experiment posts for full details and results.*

**[Scaling Follower model size and diversity](https://openreview.net/forum?id=XvCBtm5PgF&noteId=VVy5FF0Kgh).** We ran a series of new Follower experiments with larger Llama models (3B and 8B) as well as additional architectures (Qwen3-1.7B). **Key findings:**
- **DisCIPL performance scales with Follower size** (+0.09 Pass@1 with 8B vs. 1B Llama) and benefits from stronger small LMs like Qwen3-1.7B (+0.07 Pass@1 over 1B Llama).
- **DisCIPL enables better coherency/validity tradeoffs**; As Follower size scales, baselines sacrifice coherency for Pass@1; in contrast, with DisCIPL, coherency and validity are both positively correlated with Follower size.

**[Planner-only baselines: Chain-of-Thought (CoT) and Program-of-Thought (PoT)](https://openreview.net/forum?id=XvCBtm5PgF&noteId=h9DYKaqMkY).** We implemented two additional baselines that utilize the Planner in more computationally-intensive ways. **Key findings:**
- **CoT yields moderate improvements** over existing Planner-only baselines (+0.14 Pass@1); however, DisCIPL still significantly outperforms GPT-4o with CoT (+0.30 Pass@1 advantage for DisCIPL-SMC with Llama-3.1-1B).
- **PoT struggles to produce sensible text**; qualitative results suggest vanilla Python execution is not well-suited for text-generation tasks.

**[Inference cost comparisons](https://openreview.net/forum?id=XvCBtm5PgF&noteId=nkcaRtBzgM).** We present an apples-to-apples cost comparison, including detailed token usage breakdowns for each method. **Key findings:**
- **Overall, DisCIPL costs less than o1** in a direct comparison based on current API pricing (\\$4.26 vs. \\$4.75, total cost for 100 tasks).
- **DisCIPL programs are 40.1% shorter (in tokens) than o1 reasoning traces.** In addition, the tokens are generated by a cheaper model (GPT-4o vs. o1) leading to 80.2% cost savings for reasoning tokens.
- **Scalability of small Follower LMs:** The marginal cost of doubling inference-time compute in DisCIPL w/ Llama-3.2-1B (i.e., doubling the number of SMC particles) is negligible—*100x cheaper than doubling CoT length in GPT-4o* and *10000x cheaper than doubling reasoning tokens in o1*.
- **Current prompting strategy leaves significant room for efficiency optimization:** 72% of DisCIPL’s overall cost comes from the long Planner system prompt (~12.9K tokens, +/- 400 tokens); however, **95.6% of these tokens are re-used** from query-to-query. In an optimized implementation, these costs could be amortized through prefix caching or fine-tuning.

*We thank the reviewers again for the constructive suggestions that motivated these experiments.* We feel these have already significantly bolstered our work. With the remaining time in the rebuttal period, we’re happy to run any follow-up experiments or answer any questions that may arise from these findings.

*NOTE: OpenReview does not allow file uploads; key figures and supplemental data from these experiments are available at the following anonymous link:* [self-steering.notion.site](https://self-steering.notion.site/)

---

> ### Author Response · Authors · 2025-06-03
> **[NEW] Follower LM Experiments: Llama-3B, Llama-8B, Qwen3-1.7B**
>
> *NOTE: OpenReview does not allow file uploads; key figures and supplemental data from these experiments are available at the following anonymous link:* [self-steering.notion.site](https://self-steering.notion.site/)
>
> ### **Scaling up the Follower**
>
> Several reviewers (DTJV, WVvh, DG1U) commented that they would like to see evaluations of a broader range of Follower sizes to better characterize scaling effects. Accordingly, we re-ran our COLLIE experiments with multiple larger Llama-3.x models to produce 3-point scaling curves:
>
> - **1 Billion:** Llama-3.2-1B-Instruct
> - **2 Billion:** Llama-3.2-3B-Instruct
> - **8 Billion:** Llama-3.1-8B-Instruct
>
> **Key Findings**
>
> - **Pass@1 scales with Follower size:** As expected, both for DisCIPL and Follower-only baselines, Pass@1 improves with Follower model size. However, for COLLIE sentence tasks, Follower-only performance remains much lower than DisCIPL (delta Pass@1 = 0.63), even at the 8B size.
> - **Coherency improves with Follower size, but *only for DisCIPL*:** Notably, on COLLIE sentence tasks, *Coherency decreases with additional scale in the Follower-only baselines*. (See the [full results tables](https://self-steering.notion.site/#206e3927c7e180fea6a9df03e04be2d3) for Coherency numbers.)
>     - Intuitively, the larger Llama models appear to be trying harder to satisfy the constraints, producing less natural text and therefore reduced Coherency scores.
>     - In contrast, for DisCIPL-SMC/IS, we observe that Coherency and Pass@1 both *increase* with Follower size.
>
>     We believe this scaling phenomenon illustrates a unique advantage of our method over standard sampling techniques.
>
>
> **COLLIE Sentence Tasks**
>
> | Method (Llama-3.x) | 1B (Pass@1) | 3B (Pass@1) | 8B (Pass@1) |
> | --- | --- | --- | --- |
> | DisCIPL-SMC | 0.76 | **0.83** | 0.85 |
> | DisCIPL-IS | **0.84** | 0.81 | **0.88** |
> | DisCIPL-RS | 0.20 | 0.58 | 0.42 |
> | Sampling | 0.07 | 0.18 | 0.22 |
> | Beam Search | 0.08 | 0.17 | 0.22 |
>
> **COLLIE Paragraph Tasks**
>
> | Method (Llama-3.x) | 1B (Pass@1) | 3B (Pass@1) | 8B (Pass@1) |
> | --- | --- | --- | --- |
> | DisCIPL-SMC | **0.69** | 0.74 | 0.75 |
> | DisCIPL-IS | 0.66 | 0.73 | 0.71 |
> | DisCIPL-RS | 0.62 | **0.87** | **0.91** |
> | Sampling | 0.38 | 0.63 | 0.73 |
> | Beam Search | 0.38 | 0.49 | 0.63 |

---

> > ### Author Response · Authors · 2025-06-03
> > **Different Follower LM Families**
> >
> > In addition to scaling up the size of the Follower within the Llama-3.x family, we also looked at other LMs of comparable size to Llama-3.2-1B that might offer stronger performance.
> >
> > Currently, we have completed full runs of our experiments with **Qwen3-1.7B** (thanks to Reviewer WVvh for suggesting this model).
> >
> > **Key Findings:**
> >
> > - **Qwen3 indeed appears to be stronger than Llama-3.x for short-form constrained generation.** In particular, in the Follower-only (Sampling) baselines that benchmark the underlying LMs, Qwen-3-1.7B achieves similar to the larger Llama-3.1-8B (Pass@1=0.22 for both) on COLLIE sentences.
> > - **Benefits from Qwen3 also translate to DisCIPL,** where DisCIPL-SMC (Qwen-3-1.7B) outperforms the original DisCIPL-SMC (Llama-3.2-1B) by +0.07 Pass@1 on COLLIE sentences.
> > - **Qwen3’s advantages appear to be mainly on sentence-level tasks.** On COLLIE paragraph-level tasks, Qwen3-1.7B slightly underperforms Llama-3.2-1B in the Follower-only (Sampling) baselines. Accordingly, we also do not observe advantages to using Qwen3-1.7B as the Follower in DisCIPL in the paragraph setting (Pass@1=0.68 vs. Llama-3.2-1B Pass@1=0.69). It is not immediately clear to us why Qwen does not perform as well on paragraphs, but we are investigating possible explanations by comparing model outputs.
> >
> > **Qwen3-1.7B: COLLIE Sentence-Level Tasks**
> >
> > | Method | Model | Pass@1 | Coherency |
> > | --- | --- | --- | --- |
> > | DisCIPL-SMC | Qwen3-1.7B | **0.83** | 5.20 |
> > | DisCIPL-IS | Qwen3-1.7B | 0.88 | 4.76 |
> > | DisCIPL-RS | Qwen3-1.7B | 0.38 | 7.32 |
> > | Sampling | Qwen3-1.7B | 0.23 | **7.33** |
> > | Beam Search | Qwen3-1.7B | 0.24 | 6.37 |
> >
> > **Qwen3-1.7B: COLLIE Paragraph-Level Tasks**
> >
> > | Method | Model | Pass@1 | Coherency |
> > | --- | --- | --- | --- |
> > | DisCIPL-SMC | Qwen3-1.7B | **0.68** | 6.61 |
> > | DisCIPL-IS | Qwen3-1.7B | 0.65 | 5.85 |
> > | DisCIPL-RS | Qwen3-1.7B | 0.55 | 8.42 |
> > | Sampling | Qwen3-1.7B | 0.34 | **8.56** |
> > | Beam Search | Qwen3-1.7B | 0.30 | 8.52 |
> >
> > **Additional Follower LM Families: Gemma-3-1B**
> >
> > We also looked at extending these evaluations to include **Gemma-3-1B** as Follower. We ran into a small snag with vLLM compatibility (specifically pertaining to Gemma-3’s sliding window attention) that will require more time to address than we currently have due to COLM rebuttals. Nevertheless, we are excited about the possibility of including this architecture in our paper. (Thanks again to Reviewer WVvh for suggesting that we look at this model.)

---

> ### Author Response · Authors · 2025-06-03
> **[NEW] Planner Baselines: CoT and PoT**
>
> *NOTE: OpenReview does not allow file uploads; key figures and supplemental data from these experiments are available at the following anonymous link:* [self-steering.notion.site](https://self-steering.notion.site/)
>
> While our existing Planner-only baselines already evaluates GPT-4o’s performance in a direct generation setting, reviewers (e.g., 3m39) rightly point out that if we assume access to a strong Planner, there are other relevant baselines to compare to.
>
> Here, we explore two additional baselines that utilize the Planner in more computationally-intensive ways:
>
> - Planner Chain-of-Thought (CoT)
> - Planner Program-of-Thought (PoT)
>
> ### **Planner Chain-of-Thought (CoT)**
>
> We modify our existing Planner-only system prompt to include standard instructions to think step-by-step.
>
> ```
> **System Prompt**
> You are helping a user generate text that satisfies constraints. Think step-by-step in order to follow the user's instructions. Your final response should be enclosed in <answer></answer> tags.
> ```
>
> **Quantitative findings**
>
> - **On** **COLLIE Sentence-Level Tasks**, we find that CoT provides a marginal boost over GPT-4o; however, overall Pass@1 is still substantially below DisCIPL.
> - **On COLLIE Paragraph-Level Tasks,** we find that CoT does not perform standard sampling with GPT-4o.
>     - Overall Pass@1 for both methods is comparable to DisCIPL with scaled-up Followers.
>     - Notably, DisCIPL-RS (Llama-3.1-8B) still outperforms GPT-4o here (Pass@1=0.91).
>
> **COLLIE Sentence-Level Tasks**
>
> | Method | Sampling Method | Model | Pass@1 | Coherency |
> | --- | --- | --- | --- | --- |
> | Planner-only | Standard | GPT-4o | 0.32 | **8.79** |
> | Planner-only | Chain-of-Thought | GPT-4o | **0.46** | 8.73 |
>
> **COLLIE Paragraph-Level Tasks**
>
> | Method | Sampling Method | Model | Pass@1 | Coherency |
> | --- | --- | --- | --- | --- |
> | Planner-only | Standard | GPT-4o | **0.80** | **9.16** |
> | Planner-only | Chain-of-Thought | GPT-4o | 0.76 | 8.79 |
>
> **Qualitative findings**
>
> Qualitatively, we find that chain-of-thought does *not* enable GPT-4o to overcome existing issues with COLLIE tasks; for instance, counting the number of characters or positions of words.
>
> **Example #1: Character counting**
>
> ```
> **Task Prompt**
> Please generate a sentence with exactly 82 characters. Include whitespace into your character count.
>
> **GPT-4o (CoT)**
> To create a sentence with exactly 82 characters, I'll count each letter, space, and punctuation mark:
>
> "The sun sets slowly over the ocean, painting the sky with hues of orange and pink delight."
>
> Now, let's count the characters:
>
> 1. "The sun sets slowly over the ocean, painting the sky with hues of orange" (70 characters)
> 2. " and pink delight." (12 characters)
>
> Total: 70 + 12 = 82 characters.
>
> <answer>The sun sets slowly over the ocean, painting the sky with hues of orange and pink delight.</answer>
>
> **Evaluation: Invalid**
> - Length is 90 characters (expected 82)
> ```
>
> **Example #2: Word Positioning**
>
> ```
> **Task Prompt**
> Please generate a sentence:
> 1) with exactly 15 words;
> 2) with the 4th, 8th, 11th words to be 'collection', 'Noise', 'Testament' respectively.
>
> **GPT-4o (CoT)**
> To create a sentence that meets all the requirements, let's start by laying out the positions of the specific words:
>
> 1. (word 1)
> 2. (word 2)
> 3. (word 3)
> 4. collection
> 5. (word 5)
> 6. (word 6)
> 7. (word 7)
> 8. Noise
> 9. (word 9)
> 10. (word 10)
> 11. Testament
> 12. (word 12)
> 13. (word 13)
> 14. (word 14)
> 15. (word 15)
>
> Now, fill in the blanks with appropriate words to construct a coherent sentence:
>
> "The museum's vast collection included a fascinating exhibit titled Noise, featuring the Testament of artifacts."
>
> **Evaluation: Invalid**
> - word 8 is "exhibit" (expected "Noise")
> - word 11 is "featuring" (expected "Testament")
> ```

---

> > ### Author Response · Authors · 2025-06-03
> > **Planner Program-of-Thought (PoT)**
> >
> > Like DisCIPL, the Program-of-Thought **[1]** method involves asking a capable LM to generate code, which is then executed to solve the task. The key innovation in DisCIPL is that the generated code *itself* invokes one or more “Follower LMs,” whereas in Program-of-Thought, the program is a standard symbolic Python program.
> >
> > **Implementation**
> >
> > Implementing a Program-of-Thought baseline for the tasks we explore in our paper (”steering” for constrained text generation) is challenging: it is not immediately clear how standard symbolic Python programs can be used to generate satisfactory solutions to text generation tasks. We explored two different approaches:
> >
> > 1. Let the language model decide how to adapt Program-of-Thought to this domain.
> > 2. Provide in-domain examples of Python programs that solve similar tasks.
> >
> > **Qualitative findings**
> >
> > We performed evaluations for both of the above PoT approaches; the results of these explorations are documented here: [Planner Program-of-Thought Experiments](https://self-steering.notion.site/Planner-Program-of-Thought-PoT-Experiments-206e3927c7e180aa9274ec0a3eb22057).
> >
> > Overall, we found that the Python programs generated by GPT-4o with PoT generally were not well-suited for text generation; they either produced arbitrary word lists that did not make sense when joined, or failed to yield text due to various bugs. Due to the observed limitations of this approach in our setting, and given limited time in the rebuttal period, we decided not to run further PoT experiments.
> >
> > ### References
> >
> > **[1]** Wenhu Chen, Xueguang Ma, Xinyi Wang, and William W. Cohen. "Program of thoughts prompting: Disentangling computation from reasoning for numerical reasoning tasks." *arXiv preprint arXiv:2211.12588* (2022).

---

> ### Author Response · Authors · 2025-06-03
> **[NEW] Inference Cost Comparison**
>
> *NOTE: OpenReview does not allow file uploads; key figures and supplemental data from these experiments are available at the following anonymous link:* [self-steering.notion.site](https://self-steering.notion.site/)
>
> Several reviewers (DTJV, WVvh, and DG1U) asked for more detailed comparisons of the computational cost of DisCIPL with that of our baselines.
>
> For illustrative purposes, we present an in-depth study of inference costs for a representative domain (COLLIE sentence tasks). Our analysis suggests that this is a conservative choice; comparative efficiency advantages of DisCIPL are likely *even greater* on longer-form generation tasks such as COLLIE paragraph generation and PUZZLES. Detailed cost comparisons for all domains will be included in a subsequent revision.
>
> ### **Dollar cost analysis**
>
> We perform an analysis of the dollar cost of the DisCIPL-SMC method, the Follower-only, Planner-only, and o1 baselines from the submission, and the new Planner-only Chain-of-Thought baseline.
>
> **Key takeaways:**
>
> - **In absolute dollar cost, DisCIPL is cheaper than o1:** In total, running DisCIPL is marginally cheaper than o1 (\\$4.26 vs. \\$4.75 per 100 tasks).
> - **DisCIPL’s inference programs are 40.1% shorter and 80.2% cheaper to generate than o1 reasoning traces.** Programs generated by the Planner LM use 40.1% fewer tokens than reasoning traces generated by o1. In addition, the generated tokens are on average 80.2% cheaper, as they are generated by a less expensive model—GPT-4o. Overall, per 100 tasks, the tokens of generated inference programs cost \\$0.91, whereas the tokens of o1 reasoning traces cost \\$4.59.
> - **The cost of scaling inference-time compute with DisCIPL is negligible:** With 32 particles, the Follower LM’s generations account for <0.1% of the overall cost of DisCIPL. Thus, for example, doubling the test-time compute budget (e.g., from 32 to 64 particles) would have a negligible influence on the total cost of DisCIPL. By contrast, doubling the reasoning budget for o1 would nearly double its price, and similarly for Planner-only Chain-of-Thought.
> - **In the current implementation, DisCIPL’s cost is dominated by the long Planner system prompt:** Because GPT-4o has not been trained on LLaMPPL inference programs, we use in-context learning with a detailed system prompt and few-shot examples. We find that **72% of the cost of DisCIPL** results from pre-filling the part of this prompt that is static across tasks.
> - **Nevertheless, DisCIPL benefits significantly from prompt caching.** On average, **95.6% of the Planner’s prompt tokens were cached by OpenAI’s servers**, resulting in an [automatic 50% discount](https://openai.com/index/api-prompt-caching/). In a performance-optimized system, these costs could be further reduced by using a local Planner LM with prefix caching. Alternatively, we could finetune the Planner LM on inference programs—in the same way that current reasoning models are finetuned on reasoning traces—to avoid the need for expensive re-prompting. Amortizing the Planner computation in this way would yield significant practical cost savings, up to 72% of the overall cost.
>
> **Dollar Cost per 100 Tasks**
>
> |  | Answer Generation | Reasoning / Program Generation | Task Prompt | Planner System Prompt (Non-Cached) | Planner System Prompt (Cached) | Total Cost |
> | --- | --- | --- | --- | --- | --- | --- |
> | Follower-only (Llama-3.2-1B) | \\$0.0001 | - | \\$0.0003 | - | - | \\$0.0004 |
> | Planner-only (GPT-4o) | \\$0.0345 | - | \\$0.0365 | - | - | \\$0.0709 |
> | Planner-only CoT (GPT-4o) | \\$0.0376 | \\$0.2751 | \\$0.0485 | - | - | \\$0.3612 |
> | o1 | \\$0.1061 | \\$4.5887 | \\$0.0509 | - | - | \\$4.7457 |
> | DisCIPL-SMC* | \\$0.0032 | \\$0.9138 | \\$0.0003 | \\$0.2863 | \\$3.0804 | \\$4.2606 |
>
> **DisCIPL-SMC instantiated with a Llama-3.2-1B Follower, GPT-4o Planner, N=32 particles.*
>
> The columns in the above table are:
>
> - **Answer Generation:** The cost of generating the tokens of the solution to the task. For example, in o1, answer generation tokens are those generated after reasoning has completed; for GPT-4o CoT, they are the tokens tagged by the model as constituting the final answer; and in DisCIPL, they are all tokens generated as candidate answers by the Follower LM.
> - **Reasoning / Program Generation:** The cost of generating tokens instrumental to solving the task. For CoT and o1, these are [reasoning tokens](https://platform.openai.com/docs/guides/reasoning). For DisCIPL, these are the tokens of the inference program generated by the Planner LM.
> - **Task Prompt:** The cost of pre-filling the task prompt, which describes the particular task being solved. For DisCIPL, this is the cost of explaining the task to the Follower, not to the Planner.
> - **Planner System Prompt:** The cost of pre-filling the Planner’s system prompt in DisCIPL. Some of the prompt is task-specific and *not cached*, whereas most of the prompt is domain-general and *cached* across tasks.

---

> > ### Author Response · Authors · 2025-06-03
> > **Token Counts and API Pricing Details**
> >
> > The dollar costs are based on the following token counts and API prices:
> >
> > **Inference Cost for 1M Tokens** (Prices current as of 2025-05-31)
> >
> > |  | Input | Cached Input | Output |
> > | --- | --- | --- | --- |
> > | Llama-3.2-1B* | \\$0.06 | - | \\$0.06 |
> > | GPT-4o** | \\$5.00 | \\$2.50 | \\$20.00 |
> > | o1** | \\$15.00 | \\$7.50 | \\$60.00 |
> >
> > \*[https://www.together.ai/pricing](https://www.together.ai/pricing) (Note that we actually use local inference for the Follower LMs.  We use the Together API pricing for Llama-3 models with <3B parameters as a conservative cost estimate. (It’s likely that the actual cost of running Llama locally is even lower than the Together pricing.)
> >
> > \*\*[https://platform.openai.com/docs/pricing](https://platform.openai.com/docs/pricing), [https://openai.com/index/api-prompt-caching/](https://openai.com/index/api-prompt-caching/)
> >
> > **Average token counts and standard deviations per task** (per single task)
> >
> > | Method | Answer | Reasoning | Task/Follower Prompt | Planner Prompt | Planner Prompt - Cached |
> > | --- | --- | --- | --- | --- | --- |
> > | Llama-3.2-1B | 14.9 (5.3) | - | 48.3 (2.0) | - | - |
> > | GPT-4o | 17.2 (4.5) | - | 72.9 (12.3) | - | - |
> > | GPT-4o (CoT) | 18.8 (25.7) | 137.6 (83.6) | 96.9 (12.3) | - | - |
> > | o1 | 17.7 (4.2) | 764.8 (1018.3) | 33.9 (12.3) | - | - |
> > | DisCIPL-SMC* | 23.2 (11.4) per particle | 456.9 (69.4) | 48.3 (2.0) | 12894.3 (444.1) | 12321.6 (1801.8) |
> >
> > **DisCIPL-SMC instantiated with a Llama-3.2-1B Follower, GPT-4o Planner, N=32 particles.*

---

### Author Response · Authors · 2025-06-03
**Author responses to key issues**

**Plans for software release / reproducibility**

Reviewer DG1U requested clarification on reproducibility and our plans for software release:

* **LLaMPPL:** The probabilistic programming language that DisCIPL is built on is public on [GitHub](https://github.com/genlm). This library is maintained by the [GenLM Consortium](https://genlm.org/).
* **DisCIPL:** Pending the COLM review process, we are planning a GitHub release of DisCIPL with tutorials and a Docker container to aid reproducibility.

  * All experiments can be run from a single `run_benchmark.py` script with hyperparameters stored in JSON config files, to accompany the code release.
  * Timestamped LLM model strings are listed in the paper (S4.2, Experiment Setup).
  * Provided access to OpenAI API credits (see our cost breakdown) and a single GPU (A100/H100 or equivalent) for local inference, it should be possible for readers to reproduce the experiments in the paper.

Thank you for the question — we hope this clarifies our open-source plans. Please don’t hesitate to ask about any specifics you might like more information about.

**The system prompt is missing from A.10**

Due to a LaTeX bug, the system prompt (A.10) was not properly rendered in the submission PDF. (Thanks to reviewer DTJV for spotting this issue.) The relevant appendix has been corrected and will be visible in the next revision. In the meantime, for reference during the rebuttal, the system prompt has been uploaded here: **[System Prompt](https://self-steering.notion.site/Planner-System-Prompt-203e3927c7e1801e97cff7e4c12cd487)**.

**What is the role of the o1 experiments?**

In our evaluations, we included results from o1 as a representative “long chain-of-thought” reasoning model. (While o1 did not exist when we first started working on this project, we realized that reviewers would likely ask to see results from a large reasoning model for our submission.) Not surprisingly, o1 performs well on most of the tasks in our paper, and some reviewers cited this strong performance in their “reasons to reject.” Nevertheless, we believe the existing evaluation is still informative for several reasons:

* **Efficiency:** o1 is a very large model with high inference costs. We show that DisCIPL achieves near-o1-level performance with *much smaller LMs*. In particular, in our new **[Inference cost comparisons](https://openreview.net/forum?id=XvCBtm5PgF&noteId=nkcaRtBzgM)**, we show that **DisCIPL programs are 40.1% shorter (in tokens) than o1 reasoning traces.** In a direct dollar cost comparison using OpenAI’s API pricing, **DisCIPL is also less expensive than o1**.
* **Apples-to-oranges comparison:** Since our framework can be instantiated with arbitrary LMs acting as Planners and Followers, the most relevant comparison is between DisCIPL and *other models of the same size*. The strongest model used in our current instantiation of DisCIPL is GPT-4o. Therefore, the relevant apples-to-apples comparison is between DisCIPL and the **[Planner-only baselines: Chain-of-Thought (CoT) and Program-of-Thought (PoT)](https://openreview.net/forum?id=XvCBtm5PgF&noteId=h9DYKaqMkY)**, which use GPT-4o. We show that GPT-4o consistently performs worse than DisCIPL in both a direct generation and chain-of-thought setting.
* **Self-steering can also be used with large chain-of-thought models:** In principle, there is nothing preventing us from instantiating DisCIPL with o1 itself as either the Planner, Follower, or both. In practice, we face two key limitations:

  * (1) **Planner limitations for o1:** API costs make it impractical to run our experiments with o1 as a Planner (o1 is at minimum 3x more expensive than GPT-4o; that is without factoring in the cost of extended reasoning chains).
  * (2) **Follower limitations for o1:** Closed access APIs prevent us from using o1’s logprobs, which our method requires in order to compute importance weights for the Followers.

In sum, we argue that (A) our method at the current scale compares favorably in terms of both performance and efficiency *even when compared directly to a frontier reasoning model like o1*, and (B) as noted in the [COLM Reviewer Guidelines](https://colmweb.org/ReviewGuide.html), there can be value in experiments that demonstrate promise for study at a larger scale — we believe self-steering meets this description.

---

> ### Author Response · Authors · 2025-06-03
>
> **Evaluation scope is too narrow / datasets are small**
>
> Our evaluations are similar in scope to evaluations presented in other recent works on novel LLM inference methods (e.g., Tree-of-Thought **[1]**, Stream-of-Search **[2]**). In particular, **[2]** (which was an oral spotlight at COLM 2024) is a good example of a paper that introduced a novel LLM inference method and *only evaluated on a single domain* (Game of 24 / Countdown). In comparison, some of our PUZZLES tasks (e.g., budgeting, trip itinerary planning) are more representative of “real world” tasks than evaluations in prior work (e.g., simple number games, crosswords). This is not to say that we’re not interested in exploring generalization (see below), but rather that we believe the current evaluation scope meets the standard for published papers in this space while enabling us to study the properties of our method in-depth.
>
> **Can this method generalize to other tasks; e.g., math reasoning?**
>
> Some tasks in the PUZZLES domain do have quantitative reasoning components while still being, essentially, constrained generation tasks (e.g., budgeting). That said, we do believe our architecture (in which a Planner LM generates a plan, which is executed by Follower LMs) could be applied to math reasoning problems. In fact, several recent papers (**[3, 4]**, and cited in our Related Work) have shown that sequential Monte Carlo (SMC) can be used to boost performance on domains such as MATH500 and AIME. These methods typically work by scoring in-progress reasoning chains using off-the-shelf process reward models (PRMs, **[5]**); it would be interesting to explore whether, (a) DisCIPL Planners can write inference programs that effectively exploit such reward models, if given access to them; or (b) DisCIPL could generate plan programs that boost math reasoning ability *without* using special-purpose PRMs, e.g. by generating LM-as-judge prompts specialized to the problem at hand. We believe these are interesting directions for future work.

---

> > ### Author Response · Authors · 2025-06-03
> > **References**
> >
> > **[1]** Shunyu Yao, Dian Yu, Jeffrey Zhao, Izhak Shafran, Tom Griffiths, Yuan Cao, and Karthik Narasimhan. "Tree of thoughts: Deliberate problem solving with large language models." *Advances in neural information processing systems* 36 (2023): 11809-11822.
> >
> > **[2]** Kanishk Gandhi, Denise Lee, Gabriel Grand, Muxin Liu, Winson Cheng, Archit Sharma, and Noah D. Goodman. "Stream of search (SoS): Learning to search in language." *Conference on Language Models* (2024).
> >
> > **[3]** Shengyu Feng, Xiang Kong, Shuang Ma, Aonan Zhang, Dong Yin, Chong Wang, Ruoming
> > Pang, and Yiming Yang. Step-by-Step Reasoning for Math Problems via Twisted Sequen-
> > tial Monte Carlo, March 2025. URL [http://arxiv.org/abs/2410.01920](http://arxiv.org/abs/2410.01920). arXiv:2410.01920
> > [cs].
> >
> > **[4]** Isha Puri, Shivchander Sudalairaj, Guangxuan Xu, Kai Xu, and Akash Srivastava. A
> > Probabilistic Inference Approach to Inference-Time Scaling of LLMs using Particle-
> > Based Monte Carlo Methods, February 2025. URL [http://arxiv.org/abs/2502.01618](http://arxiv.org/abs/2502.01618).
> > arXiv:2502.01618 [cs]
> >
> > **[5]** Hunter Lightman, Vineet Kosaraju, Yuri Burda, Harrison Edwards, Bowen Baker, Teddy Lee, Jan Leike, John Schulman, Ilya Sutskever, and Karl Cobbe. "Let's verify step by step." In *The Twelfth International Conference on Learning Representations*. 2023.

---

### Decision · Program_Chairs · 2025-07-08

**Decision:**

Accept

**Comment:**

This is a paper on LLM inference architecture design. The architecture consists of a more powerful planner model which generates task-specific inference programs and a host of smaller executor models which implement these programs. Through this parallel design and by including techniques such as recursive search, this system can improve the cost accuracy Pareto frontier on tasks such as COLLIE, a constrained generation benchmark and PUZZLES, a dataset of tasks such as poetry, grant-writing, budgeting, and itinerary planning. It would be interesting to see what it takes to make this work on domains exact verifiers like competitive coding. The authors did a good job addressing the reviewers concerns and I'd like to recommend the paper for acceptance.